# InfoDeepSeek: Benchmarking Agentic Information Seeking for Retrieval-Augmented Generation

## Abstract

Retrieval-Augmented Generation (RAG) enhances large language models (LLMs) by grounding responses with retrieved information. As an emerging paradigm, Agentic RAG further enhances this process by introducing autonomous LLM agents into the information seeking process. However, existing benchmarks fall short in evaluating such systems, as they are confined to a *static retrieval environment with a fixed, limited corpus* and *simple queries that fail to elicit agentic behavior*. Moreover, their evaluation protocols assess information seeking effectiveness by pre-defined gold sets of documents, making them unsuitable for the open-ended and dynamic nature of real-world web environments. To bridge this gap, we present **InfoDeepSeek**, a new *benchmark with challenging questions designed for assessing agentic information seeking in real-world, dynamic web environments*. We propose a systematic methodology for constructing challenging queries satisfying the criteria of determinacy, difficulty, and diversity. Based on this, we develop the first evaluation framework tailored to dynamic agentic information seeking, including fine-grained metrics about the accuracy, utility, and compactness of information seeking outcomes. Through extensive experiments across LLMs, search engines, and question types, InfoDeepSeek reveals nuanced agent behaviors and offers actionable insights for future research.

## 1 Introduction

Despite remarkable capabilities across various domains (Lin et al., 2025; Hadi et al., 2023; Kasneci et al., 2023), large language models (LLMs) still suffer from factual hallucinations, outdated knowledge, and limited access to real-time information. To address these challenges, Retrieval-Augmented Generation (RAG) (Fan et al., 2024; Gao et al., 2023; Zhao et al., 2024) has emerged as a promising solution, enabling LLMs to enhance their responses with retrieved external information. RAG typically consists of three stages: retrieval, augmentation, and generation (Singh et al., 2025; Gao et al., 2023). The first two stages – retrieving relevant documents and selecting useful evidence – constitute the **information seeking** process. While traditional RAG systems rely on static workflows, recent advancements in Agentic RAG (Singh et al., 2025; Schneider et al., 2025; Ravuru et al., 2024; He et al., 2025) integrate autonomous LLM agents into the RAG pipeline, allowing for dynamic planning, search, and reflection to support more flexible and robust evidence acquisition. This paradigm has already been integrated into real-world systems, including Deep Research features in OpenAI (OpenAI, 2025c), Gemini (Gemini, 2025d), and Perplexity (Team, 2025a), where agents iteratively search and synthesize information from the live web.

The introduction of the agent primarily transforms the information seeking process of RAG, while the generation step remains largely unchanged, *i.e.*, responds based on the external information. Consequently, a core goal in evaluating Agentic RAG should be to assess the effectiveness of agentic information seeking. Rigorous benchmarking and evaluation are essential to quantify these improvements, identify potential weaknesses, and guide the development of more capable agentic systems. However, existing RAG benchmarks are inadequate for this purpose, as shown in Figure 1. **Firstly, most benchmarks are constrained to static environments with a fixed, limited corpus** (Yang et al., 2024; Chen et al., 2024; Lyu et al.). Such setups fail to reflect the scale and dynamic of real-world web environments, characterized by massive document volume, content drift, URL decay, and frequent

Figure 1: Comparison between traditional RAG benchmark (up) and our InfoDeepSeek (bottom).

fluctuations in search engine results. As a result, these benchmarks misalign with the operational complexity that Agentic RAG systems must manage in deployment. Moreover, static benchmarks rely on pre-defining ground-truth documents and traditional metrics such as NDCG (Yang et al., 2024). In contrast, the open-ended nature of the web makes it difficult to determine a gold evidence set in advance, rendering such metrics inapplicable. This presents a significant challenge for evaluating the quality of information seeking in dynamic environments. **Secondly, existing benchmarks often fall short in terms of question complexity.** Many of their queries are relatively simple and can be answered directly by LLMs with parametric knowledge or a single-turn search (Vu et al., 2023; Tang and Yang, 2024). Such questions fail to elicit core agentic behaviors, *e.g.*, planning, multi-turn tool use, and reasoning over multiple pieces of evidence, so they cannot meaningfully evaluate the effectiveness of agentic information seeking.

To address the above limitations, we propose **InfoDeepSeek**, a benchmark with challenging questions and novel evaluation metrics tailored for agentic information seeking under real-world web environments. First, we introduce a set of criteria and a systematic methodology for constructing challenging queries aimed at evaluating agentic information seeking. We manually curate and validate 1032 high-quality questions, each carefully designed to exhibit the following properties:

- **Determinacy**: Each question has a clear, unique, and temporally stable answer.
- **Difficulty**: The questions are intentionally challenging for LLMs, even with single-turn web search. This highlights the need for multi-turn agentic information seeking capabilities
- **Diversity**: Questions cover various domains, predominant languages, and attributes, *i.e.*, multi-hop, long-tail, freshness, time-sensitive, distracting information, and false premises.

Building on this, we develop an agentic information seeking system that integrates multiple search and browsing tools in live web environments. Facing such a noisy and dynamic environment, we propose a set of **fine-grained evaluation metrics and protocols** to dynamically assess the effectiveness of information seeking. Our evaluation metrics include answer accuracy, information accuracy, information compactness, and effective evidence utilization, offering a comprehensive view of the agent's information seeking ability. We further conduct empirical evaluations across multiple dimensions, including different LLMs, search engines, and question types, revealing agents' behaviors under complex and dynamic environments. Our key contributions are as follows:

- We introduce a set of criteria and a systematic methodology for constructing challenging queries and present a new benchmark, InfoDeepSeek, for evaluating agentic information seeking in real-world settings. We believe these principles and methodologies are transferable and can benefit the research community of benchmarking AI agents for RAG.
- We propose an Agentic RAG framework coupled with the first fine-grained evaluation metrics and protocols that assess information seeking effectiveness in dynamic environments.
- We provide a comprehensive comparison of agents under different LLMs, search engines, and question types, identifying their limitations and outlining directions for future research.

Table 1: Comparison of RAG benchmarks in factual QA. Diff.Filt. means difficulty filtering (removing questions solvable by humans or LLMs through a single-round search). Dyna.Eval. means evaluating information seeking in dynamic environments. Symbol ✗ signifies the lack of this attribute, while symbol ✗ means it is not explicitly considered.

| Benchmark | Environment | | Question | | | | | | |
|---|---|---|---|---|---|---|---|---|---|
| | Real World | Dyna. Eval. | Diff. Filt. | Multi-Hop | Long-Tail | Freshness | Time-Sensitive | Distracting Info. | False Premise |
| NQ | ✗ | ✗ | ✗ | ✗ | ✗ | ✗ | ✗ | ✗ | ✗ |
| MultiHop-RAG | ✗ | ✗ | ✗ | ✓ | ✗ | ✗ | ✓ | ✗ | ✗ |
| FreshLLM | ✓ | ✗ | ✗ | ✓ | ✗ | ✓ | ✓ | ✗ | ✓ |
| RGB | ✗ | ✗ | ✗ | ✓ | ✗ | ✗ | ✓ | ✓ | ✗ |
| CRAG | ✗ | ✗ | ✗ | ✓ | ✓ | ✗ | ✓ | ✗ | ✓ |
| BrowseComp | ✓ | ✗ | Human | ✓ | ✗ | ✗ | ✗ | ✗ | ✗ |
| BrowseComp-ZH | ✓ | ✗ | Human | ✓ | ✗ | ✗ | ✗ | ✗ | ✗ |
| InfoDeepSeek (Ours) | ✓ | ✓ | LLMs | ✓ | ✓ | ✓ | ✓ | ✓ | ✓ |

## 2 RELATED WORK

**Agentic RAG**. RAG has emerged as a key technique for enhancing the factual accuracy and timeliness of LLMs Fan et al. (2024); Gao et al. (2023); Zhao et al. (2024); Lewis et al. (2020); Salemi and Zamani (2024); Wu et al. (2024); Wang et al. (2024); Zhao et al. (2023). To overcome the limitations of traditional RAG systems – which rely on static workflows and often struggle with complex tasks (Singh et al., 2025) – the Agentic RAG paradigm has introduced agents into the RAG pipeline (Singh et al., 2025; Schneider et al., 2025; Ravuru et al., 2024; He et al., 2025; Lee et al.; Zhang et al., 2024). These agents enable multi-turn, in-depth, and dynamic information seeking, enhancing the system's performance and adaptability in complex scenarios. Notably, this paradigm has begun to see increasing adoption in practical applications, *e.g.*, Deep Research from OpenAI, Gemini, and Perplexity, all employing agents to support multi-step information seeking tasks.

**RAG Benchmarks**. Early RAG researches rely on QA benchmarks,*e.g.*, NQ (Kwiatkowski et al., 2019) and TriviaQA (Joshi et al., 2017), for evaluation. With the rapid advancement of LLMs' knowledge, recent RAG benchmarks have begun to shift to more challenging scenarios and tasks, *e.g.*, multi-source information (Yang et al., 2024), noise (Chen et al., 2024), multi-hop reasoning (Tang and Yang, 2024; Ho et al., 2020; Trivedi et al., 2022), long-tail knowledge (He et al., 2024), long document (Pradeep et al., 2025), and temporally evolving answers (Vu et al., 2023). Nevertheless, as illustrated in Table 1, most benchmarks still rely on static environments with limited corpora or limited question complexity and diversity (Yang et al., 2024; Chen et al., 2024; Tang and Yang, 2024; Kwiatkowski et al., 2019). In contrast, our work focuses on evaluating agents' information seeking abilities in dynamic, real-world settings, with challenging questions.

The evaluation of RAG involves information seeking and generation stages. Most benchmarks include assessing generation quality, *i.e.*, answer accuracy (Wei et al., 2025; Yang et al., 2024; Chen et al., 2024; Vu et al., 2023; Zhou et al., 2025). Some works evaluate information seeking quality, but they all employ retrieval metrics with pre-defined ground-truth documents (Tang and Yang, 2024; Lyu et al.; Salemi and Zamani, 2024), which is not applicable in dynamic environments without fixed ground-truth documents. Even though multi-hop retrieval datasets like BrowseComp (Wei et al., 2025) and FRAMES (Krishna et al., 2025) exist, they still only evaluate the final correct answer and lack a scalable construction method. In contrast, our approach introduces an automated construction process and a new evaluation framework for information seeking quality in dynamic settings, incorporating dimensions like relevance, utility, and compactness.

## 3 PROBLEM FORMULATION AND AGENTIC RAG FRAMEWORK

Given a user query $q \in \mathcal{Q}$, the goal of Agentic RAG is to acquire a set of evidence $C = \{c_1, c_2, \cdots, c_{n_q}\}$ of length $n_q$ by iteratively searching and browsing within an open environment, and to generate an response $\hat{y}_q$ that closely approximates the groundtrue answer $y_q$. Following the three-stage framework of RAG Singh et al. (2025), *i.e.*, retrieval, augmentation, and generation, we implement an Agentic RAG system tailored for real-world web environments. Note that we mainly focus on benchmarking the information seeking process (*i.e.*, the retrieval and augmentation stage), as it is the primary component transformed by the introduction of LLM agents into the RAG pipeline.

Table 2: Different question attributes and their ratios in our benchmark.

| Attribute | Definition | Ratio (%) |
|---|---|---|
| Multi-hop | Questions requiring chaining multiple pieces of information to compose answers (*e.g., Who directed Anne Hathaway's second film?*). | 83.82 |
| Long-tail | Questions focusing on obscure facts or entities that are hard to find on the web, e.g., a person or event about which little information is available. | 81.88 |
| Time-Sensitive | Questions involving temporal constraints with implicit/explicit time anchors (*e.g., Who was the British Prime Minister in 2013?*). | 68.31 |
| Freshness | Questions about recent (post-2025) events requiring real-time retrieval (*e.g., What is 2025 Grammy Award for Best Album?*) | 4.94 |
| Distracting Information | Search results contain significant noise, such as name ambiguity or misleading/false content (*e.g.*, fake news). | 15.41 |
| False Premise | Questions with incorrect assumptions, *e.g., How is the champion of plain high diving at 9th Olympics?* (No such event at 9th Olympics) | 4.85 |

**Retrieval Stage**. Upon receiving the input query $q$, the agent initiates a planning process $\pi_0 = \text{Plan}(q)$ about how to seek information from the web. The agent then launches an information seeking trajectory of up to $T$ steps. At each step $t$, the agent reflects on the current observation $o_t$ and its memory (*i.e.*, previous trajectory) $h_t$, and updates its plan $\pi_{t+1} = \text{Reflect}(o_t, h_t, \pi_t)$. Based on the plan, it selects tools (*e.g.*, search engines, browser, time-related utilities, or termination) and performs an action that yields the next observation: $a_{t+1} = \text{Act}(\pi_{t+1}) \to o_{t+1}$, *e.g.*, the information from web. Here we support some mainstream search engines such as Google, Bing, Yahoo, DuckDuckGo, and Selenium-based web browsing. This information seeking loop continues until the agent has sufficient information to terminate or hits the step limit $T$. This stage generates a sequence of observations $O = \{o_1, o_2, \cdots, o_T\}$, representing retrieved contents from the web.

**Augmentation Stage**. Given the potential volume and noise of retrieved content in the previous stage, the agent performs content filtering and distillation. It selects and summarizes the most relevant documents, yielding a focused set of evidence $C = \text{SelectRelevant}(q, O)$. The agent will determine the size $n_q$ of the set $C$ and sort the evidence in $C$ by importance. Usually, we only stipulate that $n_q$ does not exceed a maximum number $n$, usually $n = 5$ following previous works (Pan et al., 2023).

**Generation Stage**. Finally, the agent generates a response $\hat{y}_q$ based on the curated content $C$ and query $q$, *i.e.*, $\hat{y}_q = \text{Generate}(q, C)$. More details about our framework are provided in Appendix B

## 4 DATASET CONSTRUCTION

### 4.1 CRITERIA FOR QUERY

**Determinacy and Verifiability**. Unlike static RAG settings with a fixed corpus and information, real-world environments have constantly changing information. Thus, questions in this context must preserve stability and verifiability to allow consistent and reliable evaluation. Thus, we collect *factual questions with a clear, unambiguous, and time-invariant answer* that can be verified through publicly available web sources. This ensures robust evaluation even in dynamic environments.

**Difficulty**. If a question can be solved with LLMs' internal knowledge or LLMs with one-turn search, it fails to activate the real abilities of agents. Hence, we focus on questions that LLMs cannot answer with a single-turn search. To enforce this constraint, we apply *difficulty filtering* and exclude questions that mainstream LLMs (*e.g.*, GPT-4o (OpenAI, 2025a) and DeepSeek-R1 (DeepSeek, 2025)) can already answer correctly with a single-turn search. Furthermore, we incorporate various difficulty attributes and present their definition and ratios in our benchmark in Table 2. Note that a question can contain multiple attributes, so the sum of their ratios is not 1.

**Diversity in Attributes, Domains, and Predominant Languages**. Each query is constructed to capture a combination of at least two of the attributes in Table 2, ensuring coverage of real-world information seeking challenges. We also ensure domain diversity, including but not limited to sports, politics, science, history, geography, music, literature, art, film, gaming, and news. Besides, we consider the predominant language, cases where accurate information is more readily available in a particular language. While all questions in our dataset are provided in both English and Chinese,

we include queries whose answers are primarily documented in other languages such as Japanese, French, Korean, Italian, or Icelandic. This encourages more realistic, language-aware search behavior from the agent and creates additional challenges due to the multilingual nature of the web.

## 4.2 METHODOLOGY FOR DATASET CONSTRUCTION

To operationalize the aforementioned criteria, we develop a set of practical heuristics and workflows for query generation as shown in Figure 2. We begin by extracting knowledge from web sources, based on which we produce draft questions. These draft questions are then subjected to two key filtering stages: determinacy check and difficulty check. Questions that pass both filters are retained as candidates, and subsequently go through a multi-stage validation process. Through iterative annotation and refinement, we have developed a set of practical methodologies and guidelines that produce questions aligned with our criteria. See Appendix C for more details.

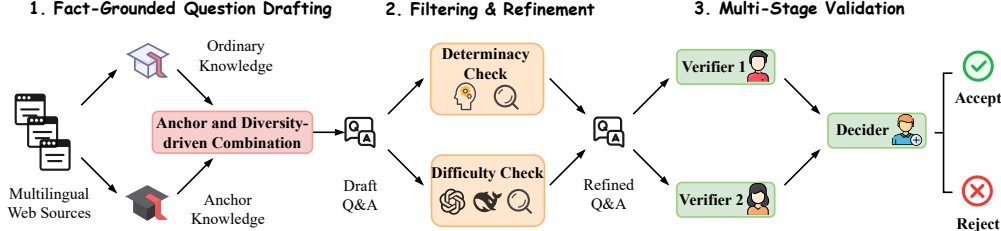

Figure 2: The construction workflow of InfoDeepSeek dataset.

**Fact-Grounded Query Drafting.** To guarantee that each question has a verifiable answer, we adopt a *reverse construction strategy* – starting from known and correct knowledge in authoritative and diverse web sources, and formulating a question with a unique answer. These sources include official websites, academic publications, or multilingual Wikipedia. This process involves two principles:

- **Expand from Anchor Knowledge.** As many complex and multi-hop questions could still be solved by LLMs through their knowledge or single search, we identify *anchor knowledge*, typically long-tail or distracting information which are hard for LLMs to answer correctly without deeper and multi-turn search. Many such anchors can be derived from low-resource or non-mainstream language sources. Another way to acquire anchor knowledge is to first find a long-tail entity and then construct relevant descriptions in the query that point to this entity. By composing these anchors with common knowledge, we create multi-hop questions that demand deeper searching and reasoning across noisy content. More detailed identification procedure of anchor knowledge can be found in Appendix C.1.1.

- **Diversification** To enhance dataset coverage, we proactively target less-common attributes, domains, or languages. Besides, starting with an anchor, we can introduce multi-hop reasoning that connects to new areas. For example, a fact about a university founder might lead to exploring their other identities to link the query to a new domain. This compositional approach systematically increases dataset complexity and diversity. More detailed instructions can be found in Appendix C.1.1.

**Determinacy and Difficulty Filtering**. In the determinacy check, each draft question undergoes *cross-referencing* against multiple independent sources to verify the correctness of the answer. This ensures that (1) the answer is uniquely correct given the query, and (2) the answer is not time-sensitive or prone to change over time. For difficulty check, we evaluate each draft question with GPT-4o and DeepSeek-R1 in a web-enabled, single-turn search setting. As LLMs have different knowledge scopes, only when a question is directly answered by both powerful models can we judge with high confidence that it is a universally simple knowledge question and it is safe to discard. This criterion ensures we maximally retain those controversial questions that are challenging and possess high discriminative power between models.

**Multi-Stage Validation for Reliability.** To ensure data quality and compliance with our criteria, each question undergoes a two-stage review process. Each query is independently verified by two

annotators, who assess its *correctness, determinacy, difficulty, and normativity*. A third adjudicator then makes the final decision regarding whether the question is eligible for inclusion.

The above construction process is highly generalizable. The first two stages can be automated by LLM agents, while the final stage can be collaboratively verified by both agents and humans. The manual construction method is detailed in Appendix C.1, and the agentic method is described in Appendix C.2. For each verified question $q$, we record its ground-truth answer $y_q$, the supporting source webpages $S_q$, and annotated metadata. In total, we collected **1032 validated data entries**, of which 360 were manually constructed and 672 were agentically constructed, covering 14 domains and 19 predominant languages. More details about data samples and data statistics are provided in Appendix C.3 and C.4.

## 5 METRICS AND EVALUATION

### 5.1 METRICS

Here, we define four metrics, assessing not only final answer accuracy but also agents' capabilities to search, extract, and prioritize relevant information from noisy sources. We denote the answer generation stage as $\phi(\cdot, \cdot)$, implemented by an LLM. See Appendix D for more details.

**Answer Accuracy (ACC)** refers to whether the answer generated based on all the observations $O$ matches the groundtrue answer $y_q$, that is $\text{ACC} = \sum_{q \in \mathcal{Q}} \mathbb{I}(\phi(q, O) = y_q)/|\mathcal{Q}|$, where $\mathbb{I}(\cdot)$ is indicator function to determine whether $\phi(q, O)$ and $y_q$ are the same, implemented by a judge LLM in Section 5.2. This is a coarse-grained correctness metric without considering augmentation stage.

**Information Accuracy (IA@k)** measures the quality of evidence obtained by information seeking process. In open web environments, predefining ground-truth documents is infeasible due to content volatility and source multiplicity, and multi-hop questions may involve different information sources. Instead, we evaluate the evidence quality by dynamically assessing whether the top-$k$ evidence of $C$ from the augmentation stage is sufficient to answer the question. Specifically, we generate an answer from the top-$k$ evidence $C_{1:k}$, *i.e.*, $\phi(q, C_{1:k})$, and compute $\text{IA@k} = \sum_{q \in \mathcal{Q}} \mathbb{I}(\phi(q, C_{1:k}) = y_q)/|\mathcal{Q}|$. A higher RA@k implies better evidence relevance. Essentially, it measures the agent's capability in the Augmentation Stage (distilling evidence from noise). A high IA@k implies the agent has successfully extracted and prioritized relevant information after retrieval.

**Effective Evidence Utilization (EEU)** measures the agent's ability to extract relevant information from the noisy observations $O$ and form the evidence set $C$. It is defined as the ratio between the best achievable accuracy across all top-$k$ subsets ($k = 1, \cdots, n$) and the answer accuracy with all observations, *i.e.*, $\text{EEU} = \frac{\max_{1 \le k \le n} \text{IA@k}}{\text{ACC}}$. EEU significantly below 1 suggests that the agent's evidence selection is suboptimal, and that key information is either buried or omitted.

**Information Compactness (IC)** quantifies the information density of evidence set $C$. An ideal agent should gather concise, high-quality evidence with minimal noise or redundancy. We first define the information compactness for each query, $\text{IC}_q$, as:

$$\text{IC}_q = \begin{cases} n_q/|S_q|, & \text{if } \exists\, k \le n_q \text{ such that } \phi(q, C_{1:k}) = y_q \\ (n+b)/|S_q|, & \text{otherwise, } \textit{i.e.,} \text{ answer failures} \end{cases}$$

where $n_q = |C|$ denotes the length of the evidence set (up to a maximum $n$), $S_q$ is the human-annotated standard set of source webpages required to answer the query, and $b$ is a penalty constant (typically $b = 1$) for answer failures. With $\text{IC}_q$, IC can be defined as $\text{IC} = \sum_{q \in \mathcal{Q}} \text{IC}_q/|\mathcal{Q}|$. IC < 1 suggests that the agent either found compact sources (covering multiple hops) or successfully leveraged prior knowledge to reduce evidence dependency. IC > 1 implies over-retrieval or poor evidence filtering with the presence of redundant or irrelevant content, even though they answer the question correctly. Note that IC does not require the agent to locate the exact same webpages as the reference set $S_q$. Here, $|S_q|$ serves as an approximate reference value, representing the inherent information complexity of the question (functionally similar to, for instance, the question's hop count). For a given test set, $|S_q|$ remains a fixed constant, ensuring that comparisons between different models are absolutely fair and consistent.

## 5.2 EVALUATION

Our proposed metrics highly rely on determining whether the LLM-generated answer, $\phi(q, C_{1:k})$ and $\phi(q, O)$, semantically and factually aligns with the groundtrue answer $y_q$. Prior work has demonstrated that LLM-based evaluators can closely approximate human judgment in factual QA (Yang et al., 2024; Xu et al., 2023). Following these findings, we adopt both human evaluation (**human-eval**) and LLM-based automatic evaluation (**auto-eval**) to assess the agreement between answers. Specifically, we mainly employ two LLM evaluators, DeepSeek-V3 and Gemini-2.0-Flash (Gemini, 2025a), to reduce self-preference bias (Panickssery et al.), following (Yang et al., 2024). If the two evaluators produce conflicting judgments, we resort to a third arbiter, GPT-4o-mini, and report the majority vote decision.

While LLM-based evaluation is generally reliable, we observe a common failure mode on false premise questions, where LLM evaluators often fail to identify incorrect assumptions in the query. To mitigate this issue, we explicitly annotate such groundtrue answers $y_q$ with statements like *"This question contains a false premise: ..."*, making the premise violation explicit. Additionally, we design separate evaluation prompts for false-premise and other questions to encourage evaluators to condition their judgment appropriately. In our experiments, this strategy improves LLMs' evaluation accuracy from 95.57% to 99.29% compared with human-eval. See Appendix D.2 for more details.

## 6 BENCHMARKING AGENTIC INFORMATION SEEKING

### 6.1 EXPERIMENT SETUP

We evaluate a range of LLMs under our Agentic RAG framework, including GPT-4o, o3-mini, Claude-3.7-Sonnet, DeepSeek-V3, DeepSeek-R1, Gemini-2.5-Flash, Gemini-2.5-Pro, and Qwen3-32B. For Qwen3-32B, we test both its thinking mode (Qwen3-32B w/ think) and non-thinking mode (Qwen3-32B w/o think). We did not consider standard RAG approaches because they typically rely on single-turn search, whereas our difficulty-filtering process explicitly excludes questions that can be answered by single-turn retrieval. Unless otherwise specified, the maximum step $T$ of retrieval stage is 5, and the maximum length of evidence set $C$ in augmentation stage is 5 ($n = 5$), as the length of supporting source webpages $S_q$ typically ranges from 1 to 3. The default search engine is DuckDuckGo, due to its open accessibility. We run the main experiment in Table 3 on the full dataset, but perform subsequent analyses on a small subset due to resource constraints. In Appendix E.7, we analyze the similarity between the subset and the full set in terms of language, attributes, and source distribution. We also compare the performance trends across different attributes to ensure that our core insights are generalizable. See Appendix E.1 for more details and analysis.

During our experiments, when evaluating a specific LLM, we use this LLM across all stages, including retrieval, augmentation, and answer generation for computing ACC and IA@k, *i.e.*, $\phi(\cdot, \cdot)$. We also explore the impact of different answer LLMs for $\phi(\cdot, \cdot)$, where information seeking and generation use different LLMs. These results are provided in Appendix E.6.

Table 3: Performance of different LLMs. ACC and IA@k are measured by %.

| Model | ACC | IA@1 | IA@2 | IA@3 | IA@4 | IA@5 | EEU | IC |
|---|---|---|---|---|---|---|---|---|
| Qwen3-32B w/o think | 5.34 | 4.32 | 4.07 | 3.94 | 3.68 | 3.81 | 0.81 | 4.59 |
| Qwen3-32B w/ think | 8.53 | 5.91 | 5.91 | 6.49 | 6.40 | 6.59 | 0.77 | 4.40 |
| Deepseek-V3 | 5.81 | 5.23 | 6.20 | 6.88 | 6.59 | 6.78 | 1.18 | 4.41 |
| Deepseek-R1 | 9.56 | 11.92 | 11.76 | 12.28 | 12.06 | 12.20 | 1.28 | 4.03 |
| GPT-4o | 7.15 | 5.70 | 6.09 | 6.38 | 6.38 | 6.47 | 0.91 | 4.42 |
| o3-mini | 10.94 | 11.82 | 11.62 | 11.82 | 12.11 | 12.11 | 1.11 | 4.17 |
| Claude-3-7-Sonnet | 12.20 | 10.80 | 11.31 | 12.33 | 12.20 | 12.20 | 1.01 | 4.50 |
| Gemini-2.5-Flash | 9.30 | 10.95 | 10.37 | 11.14 | 11.34 | 11.14 | 1.22 | 4.27 |
| Gemini-2.5-Pro | 17.73 | 22.29 | 22.67 | 22.09 | 22.19 | 22.48 | 1.28 | 4.16 |

### 6.2 BENCHMARKING ON DIFFERENT LLMS, SEARCH ENGINES, AND QUESTION ATTRIBUTES

**Different LLMs**. Table 3 presents the performance of agents based on various LLMs on our benchmark, InfoDeepSeek, highlighting the challenge it presents for agentic information seeking

tasks. **Firstly, SOTA LLMs perform suboptimally on the agentic information seeking tasks.** The best-performing model, Gemini-2.5-Pro, achieves only 17.73% on ACC and 22.48% on IA@5. This result underscores the complexity of the tasks, as even the strongest model struggles to provide accurate answers across our challenging queries. **Secondly, LLMs optimized for reasoning and information retrieval outperform others.** DeepSeek-R1 outperforms DeepSeek-V3, and O3-mini outperforms GPT-4o, indicating that reasoning models tend to perform better in agentic information seeking. Additionally, Gemini-2.5-Flash and Gemini-2.5-Pro, which are specifically optimized for search and deep research scenarios, show better performance compared to other models.

In terms of information quality (IA@$k$), most models perform poorly on IA@1, as many queries require multiple sources to provide a correct answer. A single document is often insufficient to fully address the question. As $k$ increases, we observe a trend of initial improvement followed by a decline. This is likely due to the influence of irrelevant or distracting information from later retrieved sources, highlighting the importance of effective augmentation in selecting relevant evidence. Effective Evidence Utilization (EEU) is mostly below 1, indicating that most LLMs struggle to extract useful evidence from the vast amount of information retrieved during the retrieval stage. Regarding information compactness (IC), most models exhibit significant redundancy in their responses. This is largely due to the low success rate of retrieval and the increased reliance on irrelevant information. Models with higher success rates typically exhibit lower redundancy, suggesting that reducing irrelevant evidence through better information extraction is critical for improving performance.

Table 4: Performance of DeepSeek-V3 and Gemini-2.5-Flash under different search engines.

| Model | Search Engine | ACC | IA@1 | IA@2 | IA@3 | IA@4 | IA@5 | EEU | IC |
|---|---|---|---|---|---|---|---|---|---|
| Gemini-2.5-Flash | DuckDuckGo | 14.29 | 12.65 | 15.10 | 16.73 | 16.73 | 15.92 | 1.171 | 3.750 |
| | Bing | 33.88 | 27.35 | 30.61 | 32.65 | 32.65 | 32.65 | 0.964 | 3.494 |
| | Google | **34.29** | **29.39** | **34.69** | **37.55** | **37.96** | 36.33 | 1.107 | 3.499 |
| | Yahoo | 33.47 | 28.98 | 32.24 | 35.51 | 35.10 | **36.73** | 1.098 | **3.341** |
| DeepSeek-V3 | DuckDuckGo | 8.98 | 5.71 | 7.35 | 9.39 | 9.39 | 10.20 | **1.136** | 3.926 |
| | Bing | 19.18 | 12.24 | 15.92 | 17.96 | 18.37 | 17.96 | 0.957 | 3.771 |
| | Google | 28.57 | 19.18 | 23.27 | 24.49 | 24.08 | 24.08 | 0.857 | 3.610 |
| | Yahoo | 25.71 | 17.96 | 24.08 | 26.53 | 26.94 | 26.94 | 1.048 | 3.631 |

**Different Search Engines**. To better understand the effect of different search engines on information seeking performance, we conduct controlled experiments by fixing the agent and varying the search engine. Specifically, Table 4 presents results for two representative LLMs, DeepSeek-V3 and Gemini-2.5-Flash, under four search engines: DuckDuckGo, Google, Bing, and Yahoo. **Firstly, search engine significantly affects the performance of agentic information seeking**. Google and Yahoo consistently outperform Bing and DuckDuckGo, with DuckDuckGo yielding the lowest scores. This highlights the importance of search engine quality in supporting effective agentic infromation seeking. General-purpose search engines, *e.g.*, Google and Yahoo, provide broader coverage and higher-quality results, making them better suited as information entry for Agentic RAG systems. **Secondly, a good search engine can partially compensate for model limitations**. While DeepSeek-V3 generally underperforms Gemini-2.5-Flash in information seeking tasks, its performance improves substantially when paired with Google, achieving an ACC of 28.57%, which narrows the gap with Gemini. This suggests that access to higher-quality retrieval results is especially beneficial for models with weaker reasoning capabilities. *Interestingly, EEU tends to be higher when using DuckDuckGo.* However, this may be an artifact of poor retrieval quality: when most retrieved content is irrelevant, identifying even a small number of useful pieces can lead to a higher utilization rate. This further underscores the importance of selecting strong evidence sources to support robust answer generation.

**Different Question Attributes.** To further understand where agents succeed or struggle, we analyze performance across different question attributes. Figures 3(a) and (b) show the performance of different LLMs under DuckDuckGo, while Figure 3(c) presents results of DeepSeek-V3 with different search engines. More results are available in Appendix E.2. **Firstly, LLMs and search engines consistently perform better on simpler attributes**, *e.g.*, false premise, time sensitivity, and freshness, and worse on multi-hop, long-tail, and distracting information questions. This aligns with our observations during data collection, long-tail and distracting questions often contain obscure entities, which are inherently difficult to agentic information seeking. Multi-hop questions in our benchmark are frequently compositional, often combining long-tail and distracting information, compounding their difficulty. **Secondly, reasoning-enhanced LLMs show clear advantages over base models,**

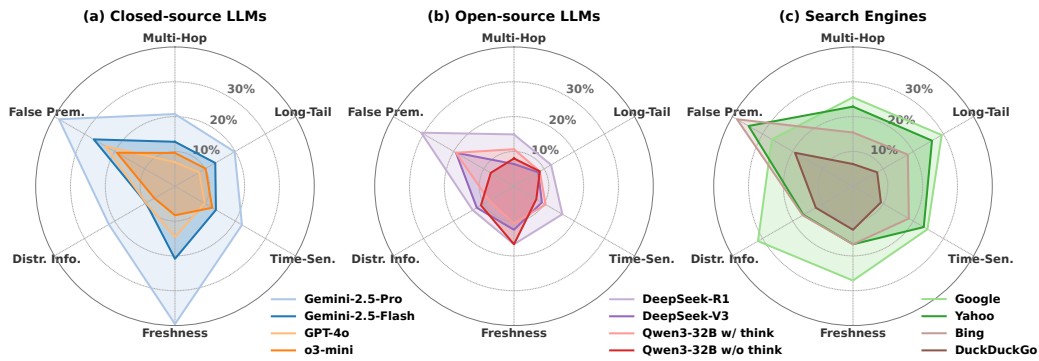

Figure 3: Performance of LLMs and search engines across different question attributes.

**but these gains are primarily observed on the simpler question attributes.** On harder attributes like multi-hop or long-tail, LLMs' (*e.g.*, DeepSeek-R1 and Gemini-Pro) performance improvements are marginal. This suggests that current LLMs, even those optimized for reasoning, are still heavily bottlenecked by retrieval quality and web information noise, particularly when facing sparse or misleading information. Lastly, Google leads to more balanced and robust performance across attributes, indicating that Google has higher information coverage and relevance. Together, these findings highlight that while LLMs and agent capabilities are essential, retrieval source quality remains a dominant factor in addressing complex information seeking tasks.

## 6.3 IN-DEPTH ANALYSIS

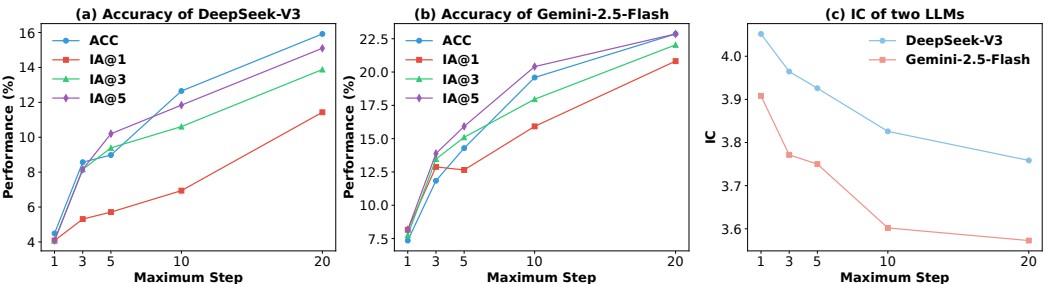

Figure 4: Performance with different maximum step $T$ of information seeking.

**Test-time Scaling for Agentic Information Seeking**. One of the key characteristics of an agent is that its performance scales with respect to the amount of compute available during test time. To investigate this, we allocate different levels of computational resources to the agent by varying the maximum step $T$ in the retrieval stage from 1 to 20, and present the results in Figure 4. As shown in the figure, both models demonstrate significant improvements in ACC, IA@k, and IC as $T$ increases, indicating clear scaling effects. This suggests that the **agent's performance can be enhanced by scaling up the test-time computing for information seeking**, with the ability to refine its search and gather more evidence as additional computation is allocated. See Appendix E.3 for more details.

**Retrieval Interference**. In our experiments, we observe a notable phenomenon where *certain questions can be answered correctly by an LLM with its parametric knowledge, but the same model fails to answer them after performing web-based retrieval*. We refer to this behavior as retrieval interference, where external information introduces confusion or distracts the model from its original correct reasoning. To quantify this effect, we define a metric called the **interference rate**, which is the fraction of questions that an LLM answers correctly without retrieval but answers incorrectly after retrieval, normalized by the total questions it initially answered correctly without retrieval. Figure 5(a) shows the interference rates of DeepSeek-V3 and Gemini-2.5-Flash across different search engines. We find that retrieval interference is widespread, suggesting that low-quality or tangentially relevant web content can often override or dilute the model's internal confidence, leading to degraded performance. To mitigate this issue, future systems should explore methods to preserve

model confidence in accurate internal knowledge and develop more precise retrieval strategies that avoid introducing misleading information. See Appendix E.4 for more results and potential solutions.

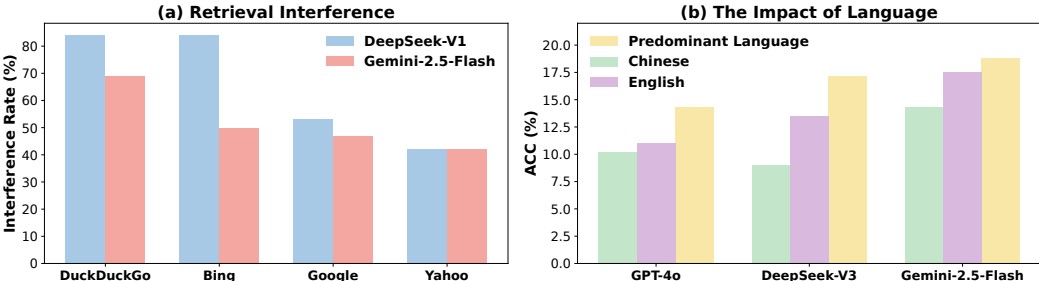

Figure 5: Retrieval interference (a) and the impact of languages (b).

**Impact of Language**. We also investigate the impact of languages on agentic information seeking process. For **Chinese** and **English**, we employ Chinese and English versions of prompts and queries. Our experiments reveal that the search keywords used by LLMs to query search tools are strongly aligned with the language of the input. For **predominant languages**, we face challenges in directly converting prompts and queries to their respective language versions. Thus, we adopt a **language-aware prompt** that explicitly instructs the agent to use the predominant language during the retrieval stage (Appendix E.5). The results in Figure 5(b) demonstrate several important trends. First, English consistently outperforms Chinese across most metrics. This is likely due to the broader coverage of English-language content and search tools. Second, predominant language prompts yield the best results. This suggests that leveraging a language-aware retrieval strategy improves the agent's ability to access and utilize high-quality, domain-relevant content.

# 7 CONCLUSION & LIMITATIONS

This work introduces InfoDeepSeek, a novel benchmark for evaluating agentic information seeking in dynamic web environments, addressing the limitations of existing benchmarks confined to a static environment and simple queries. We propose a methodology for constructing challenging queries that satisfy the criteria of determinacy, difficulty, and diversity. Furthermore, we design fine-grained evaluation metrics tailored for the comprehensive assessment of agentic information seeking under dynamic environments. In future work, we plan to explore an automated data collection approach with manual verification to lower costs and expand the dataset.

## REPRODUCIBILITY STATEMENT

To ensure the reproducibility of our work, we have made our experimental setup and data construction methodology fully transparent. The complete process for building our dataset, including detailed instructions given to the annotators, is described in Appendix C. Furthermore, we provide detailed experimental setup, including model details, hyperparameters, and computational costs, in Appendix D and E. For the convenience of review, all code and data are available at the following anonymous repository: `https://anonymous.4open.science/r/infodeepseek_code-6DA2/` and it will be made publicly available in the future. The repository includes a detailed README file, which contains clear instructions for reproducing our results.

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

TABLE OF CONTENTS

## A  THE USE OF LLMS

In this paper, Large language models (LLMs) were used as a general-purpose assist tool to improve the clarity and grammar of the manuscript. The models were not used for research ideation, data analysis, or the generation of any core content. Their role was limited to minor editing and polishing of the text to enhance readability.

## B  AGENTIC INFORMATION SEEKING FRAMEWORK

To enable complex, multi-step information seeking in open-domain environments, we design a generalizable Agentic RAG framework composed of modular components for planning, memory, tool use, and generation. The agent is instantiated around a large language model (LLM), augmented with external tools and reflection capabilities to support iterative decision-making and information seeking.

### B.1  COMPONENTS

We begin by explaining the roles of LLMs, the memory bank, and the tool library. Subsequently, these components will be integrated into the primary agent loop.

**LLMs**. The LLM serves as the agent's central reasoning engine. It is responsible for interpreting the user query, generating search plans, selecting tools, reflecting on past actions, filtering retrieved content, and synthesizing the final answer. Our framework supports a variety of LLMs, including both API-accessible remote models and locally hosted open-source models. To accommodate the varying requirements of different LLMs, we introduce a straightforward API calling that accepts a prompt as input and returns a response. All agents follow a unified interface that incorporates structured planning and reflection mechanisms: at each step, the agent is prompted to explicitly plan its next action and reflect on prior steps to refine its strategy. This iterative planning-reflection loop enhances the agent's adaptability to noisy or ambiguous web content.

**Memory**. The memory stores the evolving trajectory of the agent's interaction with the web environment. Concretely, it includes (1) all past plans generated by the agent, (2) actions such as tool invocations and search queries, and (3) the corresponding observations retrieved from the web (e.g., snippets, page titles, contents). This memory is continuously updated and used as input to future planning and reflection steps, enabling the agent to reason over previously collected evidence, avoid redundancy, and refine its search strategy over time.

**Tool Library**. Tools serve as the agent's interface to the external world. For our information seeking agent, we support multiple real-time search engines, Google, Bing, Yahoo, and DuckDuckGo, as well as a Selenium-based browser tool that allows the agent to navigate, scroll, and extract content from live webpages. In addition, we support time tool for time calculation that may be involved in queries and webpages. This diverse toolset ensures robustness across query types and web content structures.

### B.2  RETRIEVAL STAGE

The Retrieval Stage is responsible for actively exploring the web environment. The agent interacts with external tools (e.g., search engines, browsers) through multi-step planning and decision-making to acquire potentially relevant information. This stage emphasizes dynamic behavior: the agent iteratively queries, observes, reflects, and adapts its strategy based on the evolving context.

Upon receiving the input query $q$ from the user, the agent initiates an initial planning process $\pi_0 = \text{Plan}(q)$, which specifies its strategy for acquiring relevant information from the web. The agent then launches an information seeking trajectory consisting of up to $T$ iterative steps. At each step $t$, it receives an observation $o_t$ from the environment (*e.g.*, search results or web content), and updates its internal plan by reflecting on the current observation $o_t$ and memory (*i.e.*, previous trajectory) $h_t$:

$$\pi_{t+1} = \text{Reflect}(o_t, h_t, \pi_t)$$

Based on the updated plan $\pi_{t+1}$, the agent selects the next action using a tool from its available set – such as a search engine (Google, Bing, Yahoo, DuckDuckGo), a Selenium-based browser for

webpage exploration, or auxiliary tools like a time utility or stop action:

$$a_{t+1} = \text{Act}(\pi_{t+1}) \rightarrow o_{t+1}$$

Each action yields an observation $q_{t+1}$, typically the results of the tool, *e.g.*, a snippet, webpage content, or search result. This planning-action-reflection loop allows the agent to dynamically adapt its strategy in response to retrieved evidence. The loop terminates either when the agent deems the collected information sufficient or when the step limit $T$ is reached. The output of this stage is a sequence of retrieved observations $O = \{o_1, o_2, \cdots, o_T\}$, representing the raw web content gathered during the search process. The detailed prompt of the retrieval stage is listed as follows:

> You are a {agent_name}, {agent_bio}, {agent_instructions}
>
> Currently, you are in the task planning phase, where you will be given a specific query to address. Please utilize LLM's advantages and pursue efficient strategies for task planning.
>
> 1. You have a short-term memory of approximately 4,000 characters.
>
> 2. You do not require assistance or response from users.
>
> 3. You can use the reference tools mentioned when planning.
>
> 4. Complex problems can be split into sub-problems and then information can be collected, aggregated and authenticated. Be sure to verify the truthfulness of the information.
>
> 5. Stay humble and call the tool for questions you are not sure about, but do not call the same tool with the same parameters repeatedly.
>
> 6. You can think and plan up to {max_iter_num} steps, so strive to plan tasks as efficiently as possible.
>
> 7. You have the capability for reflection and self-criticism; reflect on past decisions and improve your strategies.
>
> 8. If you have sufficient information to answer the given query, invoke the termination tool to terminate planning. Otherwise, continue planning new tasks while ensuring no duplication with prior tasks.
>
> {tool_specification}
>
> {current_date_and_time}
>
> {memory}
>
> Given Query: {query}
>
> Based on the given question and existing tasks, plan a new Task (no repetitions), and you can only generate the Task in the following **JSON list** format:
>
> ```
> [{
>     "task_name": "task description",
>     "command":{
>         "name":"command name",
>         "args":{
>             "arg name":"value"
>         }
>     }
> }]
> ```
>
> Even if there is only one task or no task, it needs to be returned in the form of a list. Ensure that the Task can be parsed by Python's json.loads function.
>
> If the completed Tasks are sufficient to answer the query, terminate planning. Otherwise, create another Task that does not duplicate previous ones.
>
> A new Task:

### B.3 AUGMENTATION STAGE

The Augmentation Stage focuses on filtering and organizing the retrieved content. Since web data is often noisy, redundant, or only partially relevant, this stage distills the raw observations into a compact, high-quality evidence set. It ensures that only the most pertinent information is retained for answer generation, improving factual grounding and mitigating hallucinations.

Given the potentially large and noisy set of retrieved content $O$, the agent proceeds to filter and distill this information into a more concise, relevant set. Specifically, it applies a selection function to identify passages or documents that are most pertinent to answering the query:

$$C = \text{SelectRelevant}(q, O).$$

This process includes both document-level and span-level selection, ranking evidence based on relevance, coverage, and redundancy. The resulting set $C = \{c_1, c_2, \cdot, c_{n_q}\}$ is sorted by importance, where $n_q$ is determined dynamically by the agent, subject to an upper bound $n$. This stage is critical for reducing noise and focusing generation on high-quality information. The prompt for the augmentation stage is as follows:

```
You are a {agent_name}, {agent_bio}, {agent_instructions}
The current stage is webpage ranking stage.  In the previous
interactions, you have already found several webpages in
response to the user's query.  Now, you need to consolidate
this information and select the {max_webpage_num} most
relevant webpages, then rank them.
1.  A webpage consists of a URL and webpage summary or
information extracted from the webpage that is relevant to
the query.
2.  If multiple pieces of information come from the same
webpage (determined by identical URLs), merge them rather
than listing duplicates.
3.  The output webpage list must include relevant webpages
necessary to answer the question.  If the question has
multiple sub-questions, the relevant webpages of each
sub-question must be included.
4.  The number of webpages in the output webpage list
can be less than {max_webpage_num}.  If it is more than
{max_webpage_num}, select {max_webpage_num} of the most
important ones.
5.  The output webpage list is sorted according to its
importance to answering the question, that is, the webpage
ranked first has the greatest contribution to answering the
question.
{current_date_and_time}
{memory}
Given Query:  {query}
You must generate the list of webpages strictly in the
following **JSON list** format:

[{
    "url": "The webpage's URL",
    "content": "Information extracted from the webpage that is
    ↪   relevant to the given query",
}]

Always return a list, even if there is no relevant web page,
you need to return an empty list to ensure that the task can
be parsed by Python's json.loads
Relevant webpages (ranked by importance):
```

### B.4 GENERATION STAGE

The Generation Stage uses the refined evidence to produce a final response. Grounded in the selected content, the language model synthesizes an answer that directly addresses the original query, ideally with high factual accuracy and traceability. In the final stage, the agent generates an answer $\hat{y}_q$ based on the curated evidence set $C$ and the original query $q$. The generation function is typically a forward pass through the LLM, grounded in the selected content:

$$\hat{y}_q = \text{Generate}(C, q).$$

This answer reflects the agent's ability to synthesize multiple sources of retrieved knowledge and produce a coherent, factually accurate response.

```
You are {agent_name}, {agent_bio}, {agent_instructions}
Currently, you are in the question-answering stage.  Based
on your own knowledge and relevant webpages, answer the given
query from the user.
1.  If the user's query contains multiple answers, list all of
them.
2.  If the user's query is based on a wrong premise, point out
the error.
{current_date_and_time}
Given query:  {query}
Relevant webpages:  {webpages}
Generate a brief English answer to solve the user's query:
```

## C DATASET CONSTRUCTION

### C.1 MANUAL CONSTRUCTION

Our query construction pipeline consists of four key stages: draft generation, validation, annotation, and verification. This section outlines the practical methodology we adopt to ensure that the final queries are factually grounded, sufficiently challenging for LLMs, and diverse across difficulty attributes, domains, and languages.

#### C.1.1 DRAFT QUESTION GENERATION.

We adopt a *reverse construction strategy*, where factual knowledge is first extracted from credible sources and then transformed into a question that requires that knowledge (and possibly more) to answer. The goal is to create queries that are multi-hop, complex, and grounded in long-tail or noisy information. The main process proceeds as follows:

1. **Anchor knowledge identification**. Annotators begin by selecting an underrepresented domain or language. They search authoritative sources such as Wikipedia, reputable news sites, fact-checking platforms, expert forums, and academic databases to extract candidate facts—particularly long-tail or distracting information, which we refer to as anchor knowledge. Wikipedia's link structure, category graphs, and multilingual variants are especially helpful for uncovering obscure entities or facts. Some webpages present different content across language versions; for example, Wikipedia entries in certain "advantage languages" often contain more detailed information. Knowledge unique to low-resource or non-English versions is more likely to be long-tail and underrepresented elsewhere.

2. **Question composition**. Based on the anchor knowledge, annotators either directly construct a question or further increase question difficulty by incorporating multiple challenge attributes or introducing additional domain-specific facts – either common or difficult – through multi-hop composition. Compositionality is encouraged to ensure that the question cannot be solved with shallow retrieval.

3. **Temporal stability check**. Annotators have to verify whether the answer varies with time. For potentially unstable questions (e.g., "Who is the current president of the USA?"), we explicitly add time constraints (e.g., "In 2025") to ensure the answer remains fixed and verifiable.

4. **Diversity control**. We aim for each question to include at least two difficulty attributes, and for the dataset to span a wide range of domains, languages, and countries. Annotators actively switch focus when certain attributes or domains become overrepresented.

**Clear Definition and Identification Procedure for Anchor Knowledge.** In our work, "Anchor Knowledge" serves as the starting point for our "Reverse Construction Strategy" (constructing questions and answers backward from a text). It is operationally defined as information that is either Long-tail or Distracting.

For Long-tail Knowledge, we have:

- **Operational Definition**: Knowledge with low frequency, typically where answers are hard to find directly on the first page of search results or where LLMs fail to recall (hallucinate), often derived from unique knowledge existing "only in low-resource or niche language webpages".

- **Identification Procedure (Manual)**: When navigating various webpages, annotators are explicitly guided to actively look for knowledge fitting the long-tail definition. Criteria include: whether the page has low traffic; whether the knowledge exists primarily in niche language pages; whether the knowledge is hard to explicitly obtain after searching; and whether SOTA LLMs (e.g., GPT-4o) struggle to answer it directly.

- **Identification Procedure (Automated)**: We utilize Wikidata's SPARQL service to construct a knowledge graph. Operationally, we define "nodes with fewer than 10 associated entities" as Long-tail nodes. We then use the documents and knowledge associated with these nodes as the starting point for constructing questions.

For Distracting Information, we have:

- **Operational Definition**: Knowledge that easily introduces noise during retrieval. As our automated pipeline did not focus on this specific category, we provide the manual identification scheme below.

- **Identification Procedure (Manual)**: Annotators are guided to actively seek cases that are prone to introducing noise. Criteria include: the existence of name ambiguity or confusingly similar entities; whether the entity is associated with fake news; and whether conventional search returns a large volume of irrelevant but plausible distracting entities.

**Systematic Rules for Diversification.** Our diversity is actively controlled through two clear, non-arbitrary rules:

- **Active Starting Point Selection**: At the very beginning of the construction process, our guidelines explicitly require the annotator or Agent to "begin by selecting an underrepresented domain or language". This selection is not "arbitrary" but determined based on real-time statistical analysis of the domain and language distribution of the data already collected.

- **Active Process Control and Correction**: During construction, we enforce a specific "Diversity Control" step. Annotators or Agents are required to execute a non-arbitrary correction procedure: "Annotators actively switch focus when certain attributes or domains become overrepresented".

### C.1.2 FILTERING AND REFINEMENT.

Each draft question undergoes a two-stage validation process to ensure that it meets both difficulty and determinacy standards.

- **Difficulty check**. To ensure that a question cannot be answered by an LLM with internal knowledge or single-turn search, we test it with GPT-4o and DeepSeek-R1 with web access enabled. If both models correctly answer the question in a single turn, the question is discarded.

- **Determinacy check**. We verify the correctness and uniqueness of the answer using multi-source cross-validation: (a) Searching and confirming the answer through multiple independent web sources. (b) Leveraging links and citations provided by web-enabled LLMs during difficulty filtering to trace and confirm factual accuracy. (c) Comparing content across different language versions of the same source (e.g., multilingual Wikipedia pages) to check for consistency and factual reliability.

### C.1.3   QUESTION ANNOTATION.

For each validated question, we record:

- The Chinese and English versions of question $q$.
- The Chinese and English versions of ground-truth answer $y_q$.
- The set of source webpages $S_q$ that provide the factual basis for the answer.
- Annotated metadata including difficulty attributes, domains, and predominant languages.

For multi-hop questions, we annotate the answer and evidence source for each reasoning step. For false premise questions, we explicitly mark the flawed assumption in the answer using the phrase: "This question contains a false premise: ...", a format critical for subsequent automatic evaluation. In addition, special attention should be paid to the translation of proper nouns, which should be accurately translated based on online sources or annotated with their original names in the predominant language.

### C.1.4   MULTI-STAGE VALIDATION

**Validation from Two Verifiers.**   To ensure the reliability and robustness of each constructed question, we implement a multi-stage human verification process involving two independent annotators (verifiers). Each verifier is required to evaluate the question from multiple perspectives. The validation process includes:

1. **Content Verification**: Check the correctness of the question and answer, and ensure that the listed sources support the answer. Special attention is paid to the accuracy of proper noun translations, which must be verified against online references or annotated using their original names in the predominant language.

2. **Criteria Check**: Evaluate whether the question meets the required conditions: Determinacy (Is the answer stable over time?) and Difficulty (Can GPT-4o and DeepSeek-R1 in a web-enabled, single-turn search model correctly answer the question?) Regardless of whether the question passes, the LLM responses are recorded in the validation notes for reference during final review.

3. **Metadata Verification**: Ensure the correctness of annotated attributes, domain, and advantage language.

4. **Validation Outcome**: Verifiers must fill out three fields: their name, a binary result (pass/fail), and explanatory notes. A question is marked as passed only if it fully satisfies all requirements (content, criteria, metadata). A fail may indicate outright rejection or suggest that further review or correction is needed.

**Final Decision**   An additional decider makes the final decision based on the verification results from the two verifiers. If both verifiers mark the question as valid, it is accepted directly. If either verifier marks it as invalid, a third annotator conducts a further review, discarding questions that do not meet essential criteria and correcting others where appropriate. This includes the following cases:

- Incorrect metadata: Fix and accept.
- Time-dependent Answer: Add a time constraint and re-validate with LLMs.

- Inaccurate answer: Replace with a correct one (confirmed by LLMs and sources) and accept.

- Ill-formed or ambiguous question: Reject.

- Inconsistent difficulty judgments: Since LLM behavior can vary, we accept questions if at least three out of six LLMs (by the collector, two verifiers, each using two LLMs for difficulty check) result in incorrect answers.

## C.2 Agentic Construction

Since the problem construction process we proposed in the paper is generalized, it can be done manually or automated using LLM agents. Thus, We present how we use agents to automate the construction workflow proposed in our paper to reduce costs and reliance on manual annotation. This agent-based dataset construction adheres to the dataset construction workflow we proposed in our paper, only with agents performing most of the steps.

### C.2.1 Fact-Grounded Question Drafting (by Agent)

This stage follows the original workflow by extracting anchor and ordinary knowledge from web documents to draft questions — now fully handled by the agent. First, given an anchor entity from Wikidata's SPARQL service, the agent performs web search and browsing to retrieve entities related to the anchor entity and their relationships (a small-scale knowledge graph), as well as documents relevant to the entities. Then, based on the retrieved knowledge graph and supporting documents, the agent summarizes anchor and ordinary knowledge. Finally, the agent composes questions based on both types of knowledge. Note that both knowledge summarization and question generation are guided by a set of seed question patterns summarized from existing human-curated data.

### C.2.2 Filtering and Refinement (by Agent)

This agent would automate the determinacy and difficulty checks in our original workflow. For determinacy, it would verify answer correctness, time-invariance, and ambiguity against source documents. For difficulty, it would invoke external LLMs and search tools to answer the generated questions, then judge if the question passes the difficulty criteria.

### C.2.3 Multi-Stage Validation (Agent-Human Collaboration)

This stage continues to follow the multi-stage validation strategy proposed in the paper, with the first-stage verification now performed by an agent instead of a human.

1. **First-Stage Verification (by Agent)**: Two additional agents, based on different backbone LLMs, independently verify the question and answer against our defined standards, i.e., whether the answer is correct and deterministic, and whether the question is difficult.

2. **Final-Stage Decision (by Human)**: Human annotators would perform a final check to ensure reliability and avoid hallucinations. This validation is necessary and significantly less costly and time-consuming than full manual construction, providing efficient, high-quality control.

To ensure question quality, we invested substantial manual effort throughout the collection and validation process. In total, seven annotators were involved in drafting questions, each of which was further reviewed by two independent verifiers and a final decider. Due to the complexity, cross-validation, and manual nature, we ultimately curated 1032 high-quality questions, of which 360 were manually constructed and 672 were agentically constructed.

## C.3 Data Examples

Here, we provide two examples of our dataset. The first question exemplifies a challenging combination of distracting information, long-tail knowledge, and multi-hop reasoning. To answer it, one must first identify the president in question and then determine which university awarded him a master's degree, making it a classic multi-hop task. The query's mention of "a president who is a comedian" naturally points to Ukrainian President Volodymyr Zelensky,

and many search results indeed return Zelensky-related pages, demonstrating the presence of strong distractors. However, the actual target is Jimmy Morales, former president of Guatemala. His English Wikipedia entry (`https://en.wikipedia.org/wiki/Jimmy_Morales`) only notes that he holds a master's degree in strategic studies with a specialization in security and defense from Mariano Gálvez University. Only by consulting less-common Spanish-language sources (for example, an archived page at `https://web.archive.org/web/20151028180235/http://www.jimmymorales.gt/main01/?page_id=2`) can one discover that he also earned "a master's degree in media management from Universidad Panamericana de Guatemala." This obscure detail represents the long-tail information necessary to fully answer the question.

```
{
    "id": 307,
    "determinacy": true,
    "difficulty_GPT": true,
    "difficulty_DS": true,
    "multi_hop": true,
    "long_tail": true,
    "time_sensitive": false,
    "freshness": false,
    "distracting_info": true,
    "false_premise": false,
    "domain": [
      "politics",
      "education"
    ],
    "advantage_language": [
      "Spanish"
    ],
    "query_zh": "某个国家的第50任总统同时是一位喜剧演员，他在哪里获得硕士学
    ↪  位？",
    "answer_zh": "这位总统是危地马拉第50任总统吉米·莫拉莱斯（Jimmy Morales
    ↪  ），他从马里亚诺·加尔韦斯大学（Mariano Gálvez University）获得高级战
    ↪  略研究硕士学位，从危地马拉泛美大学（Universidad Panamericana de
    ↪  Guatemala）获得媒体管理硕士学位",
    "query_en": "The 50th president of a certain country is also a comedian.
    ↪  Where did he get his master's degree?",
    "answer_en": "The president is Jimmy Morales, the 50th president of
    ↪  Guatemala. He holds a master's degree in strategic studies with a
    ↪  specialization in security and defense from Mariano Gálvez
    ↪  University and a master's degree in media management from
    ↪  Universidad Panamericana de Guatemala.",
    "sources": [
      "https://en.wikipedia.org/wiki/Jimmy_Morales",

      ↪  "https://web.archive.org/web/20151028180235/http://www.jimmymorales.gt/main01/?page_id=
    ],
    "verified": true
  }
```

The second question illustrates a false-premise challenge. It asks: "Which composer was awarded to the 2022 Grammy Awards Songwriter of the Year?" The underlying premise – that the Grammy Award for Composer of the Year existed in 2022 – is false. In reality, this category was only established in 2023. During the search, one will find references to the 2023 recipient, creating additional noise and demonstrating how real-world information seeking can be misled by a question's incorrect assumptions.

```
  {
    "id": 209,
    "determinacy": true,
```

```
    "difficulty_GPT": false,
    "difficulty_DS": true,
    "multi_hop": false,
    "long_tail": true,
    "time_sensitive": true,
    "freshness": false,
    "distracting_info": true,
    "false_premise": true,
    "domain": [
      "entertainment"
    ],
    "advantage_language": [
      "English"
    ],
    "query_zh": "2022年格莱美奖年度作曲人颁给了哪位作曲家",
    "answer_zh": "格莱美奖的年度作曲人（非古典）奖（Songwriter of the Year,
→   Non-Classical）首创于2023年，隔年被正式加入通类。所以该问题存在虚假前
→   提，即22年没有这个奖",
    "query_en": "Which composer was awarded to the 2022 Grammy Awards
→   Songwriter of the Year?",
    "answer_en": "Grammy Award fo Songwriter of the Year (Non-Classical)
→   was first presented in 2023 and was not previously part of any
→   specific field. Therefore, this question has a false premise, that
→   is, there is no such award in 2022.",
    "sources": [
      "https://zh.wikipedia.org/wiki/%E8%91%9B%E8%90%8A%E7%BE%8E%E7%8D%8E"
    ],
    "verified": true
  }
```

## C.4 DATA STATISTICS

After rigorous drafting, refinement, and multi-stage verification by seven annotators, our dataset comprises 1032 high-quality queries, each exhibiting at least two of the following six difficulty attributes: Multi-Hop, Long-Tail, Time-Sensitivity, Freshness, Distracting Information, False Premise. Their ratios in our benchmark are shown in Table 2

We also ensured domain diversity by encouraging annotators to cover a wide range of topics. In total, queries span 14 broad domains, including history, geography, film & TV, science & technology, literature & art, politics, education, music, news, sports, humanities, entertainment, games, and social sciences. For visualization, closely related subfields (e.g., astronomy, biology, medicine, and computer science) are grouped under "science & technology," while economics, sociology, and law fall under "social sciences." The question ratio of each domain is presented in Figure 6; since multi-hop questions may touch multiple domains, the percentages sum to more than 100%.

Finally, we annotated each query with its predominant language, the language in which relevant evidence is most readily available. While English and Chinese are dominate, reflecting the abundance of resources in these languages, our dataset also includes 17 less common predominant languages: Japanese (11), Russian (9), Korean (8), Italian (6), Arabic (6), French (5), Spanish (4), German (3), Portuguese (3), Icelandic (3), Slovene (3), Malay (2), Bengali (2), Croatian (1), Czech (1), Dutch (1), and Hindi (1). This linguistic variety ensures that our benchmark challenges agents to perform cross-lingual retrieval and to leverage non-English sources when appropriate, further broadening the realism and difficulty of the tasks.

## D EVALUATION

To ensure reliable assessment of agent-generated answers, we adopt both human evaluation (**human-eval**) and automatic evaluation (**auto-eval**) using LLMs, each with carefully designed guidelines to handle diverse question types, including those with false premises.

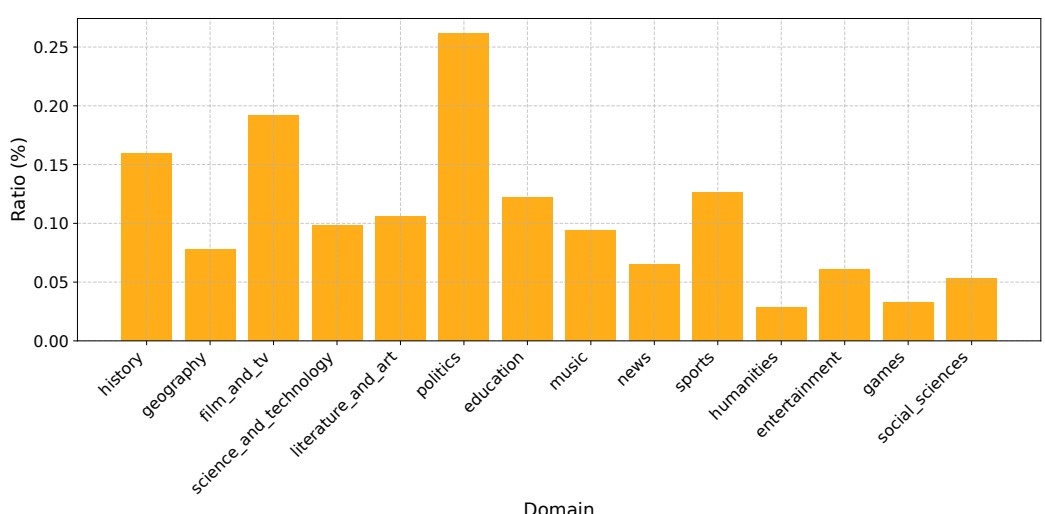

Figure 6: The ratio of questions for different domains.

## D.1 HUMAN EVALUATION

Human annotators are asked to determine whether the agent's answer $\hat{y}_q$ correctly answers the given question, with respect to the ground-truth answer $y_q$. The evaluation guidelines emphasize the following key aspects:

- **Entity consistency**: Ensure that events, people, and locations mentioned in the answer match the ground truth.
- **False premise detection**: If the question contains a false assumption, the answer must explicitly point it out.
- **Answer completeness**: If the question requires multiple pieces of information, all must be answered correctly.

Each instance is independently annotated by two human raters, and they reach a high agreement level of 97%, indicating strong reliability. In cases of disagreement, a third expert annotator adjudicates the final decision. For a new model, we randomly sampled 10% of the data for manual review. Specifically for the experiments presented in Table 5, we evaluated 103 samples per model.

## D.2 AUTOMATIC EVALUATION

We primarily employ two strong LLM evaluators, DeepSeek-V3 (`deepseek-chat`) (DeepSeek, 2025) and Gemini-2.0-Flash (`gemini-2.0-flash`) (Gemini, 2025a), to mitigate model-specific bias and self-preference, following the approach proposed in (Yang et al., 2024). If the two models produce conflicting judgments, we resort to a third evaluator, either GPT-4o-mini (`gpt-4o-mini-2024-07-18`) (OpenAI, 2025a)) or a human judge, and take the majority vote as the final answer correctness label.

Initially, we employed a **single unified prompt** to evaluate all questions, regardless of type. The general-purpose prompt template was as follows:

```
Given a question and its groundtrue answer, determine whether
the candidate answer is consistent with the correct answer.
Note the following points:
1.  The candidate answer must contain content consistent with
groundtrue answer to be considered correct (pay attention to
the consistency of time, place, and person), but more details
can be provided.
```

Table 5: The Accuracy of unified prompt and separate prompt strategies in automatic evaluation compared to human evaluation.

| | DeepSeek-V3 | | Gemini-2.0-Flash | |
|---|---|---|---|---|
| | unified prompt | separate prompt | unified prompt | separate prompt |
| **Accuracy** | 95.57% | 99.29% | 96.43% | 98.71% |

```
2.  If there are multiple items in the groundtrue answer,
the candidate answer must also contain all the items to be
correct.
3.  If the groundtrue answer mentions that the premise of the
question is wrong, such as some assumptions of the question
are wrong, the candidate answer must point out the wrong
premise to be considered correct.
4.  When the groundtrue answer does not mention the wrong
premise, the question itself is not problematic.  If the
candidate answer says that the question has a wrong premise,,
such as person/event may not exist, it is a wrong answer.
5.  The given question has a clear answer and the given
groundtrue answer is absolutely right.  If the candidate
answer says it doesn't know or lacks information, it is a
wrong answer
Just answer Yes or No.
Given Question:  {query}
Groundtrue Answer:  {reference_answer}
Candidate Answer:  {candidate_answer}
Is the candidate answer correct?
```

As shown in Table 5, this single unified prompting strategy achieves an overall accuracy of 96% when compared to human evaluation. However, we identified a consistent failure mode on false premise questions, where LLM evaluators often fail to recognize implicit factual errors or assumptions in the query (see Section D.3 for examples).

To mitigate this issue, we explicitly annotate such groundtruth answers $y_q$ with phrases like: *"This question contains a false premise: ..."*, making the logical error explicit. Furthermore, we develop a **separate-prompt strategy**, with one prompt tailored for false-premise questions and another specifically for other questions. The specialized prompt for false-premise questions is as follows:

```
Given a question and its ground-truth answer, determine
whether the candidate answer correctly answers the given
question.  Pay attention to the following points:
1.  This question has a false premise, which has been pointed
out in the groundtruth answer.  If the candidate answer does
not point out or correct this false premise, it is incorrect.
2.  If the false premise pointed out by the candidate answer
is different from the groundtruth answer (time, place, event,
person, meaning inconsistent), it is incorrect.
3.  If the groundtruth answer still answers the question in
addition to pointing out the false premise, the candidate
answer should also answer the question.  In this case, if the
candidate answer does not answer the question or the meaning
of the answer content is inconsistent with the groundtruth
answer (pay attention to the consistency of time, place,
person, and quantity), it is incorrect.
Just answer Yes or No.
```

```
Given question: {query}
Groundtruth answer: {reference_answer}
Candidate answer: {candidate_answer}
Does the candidate answer correctly answer the given question?
```

The prompt for other questions is as follows:

```
Given a question and its groundtrue answer, determine whether
the candidate answer correctly answers the given question.
Pay attention to the following points:
1.  The candidate answer must contain content that is
consistent with the groundtrue answer to be considered correct
(pay attention to the consistency of time, place, person, and
quantity), but more details can be provided.
2.  If there are multiple contents/events/persons in the
groundtrue answer, the candidate answer must also contain
all the contents/events/persons to be considered correct.
3.  The given question does not have a wrong premise, and
the relevant person/event must exist and be unique.  If the
candidate answer proposes a wrong premise or cannot determine
whether the person/event exists, it is a wrong answer.
4.  The given question has a clear answer and the given
groundtrue answer must be correct.  If the candidate answer
does not answer the question correctly but proposes the need
to further query relevant information, it is a wrong answer.
Just answer Yes or No.
Given question: {query}
Groundtruth answer: {reference_answer}
Candidate answer: {candidate_answer}
Does the candidate answer correctly answer the given question?
```

As shown in Table 5, this separate-prompt strategy significantly improves evaluation robustness, increasing LLM evaluation accuracy from 95.57% to 99.29% compared to human annotations.

Regarding the concern that "harder questions might be harder to evaluate," we believe this can be mitigated in two ways:

- First, false premise questions are a highly specialized type, possessing a unique answer structure due to the requirement to explicitly point out the error. We believe such structurally distinct types are a minority in the dataset. Scaling to new difficulty attributes in the future does not necessarily imply a proliferation of similar, hard-to-evaluate structures.
- Second, even if new difficult-to-evaluate types emerge, our approach to handling false premises provides a scalable template. We can replicate this success: utilizing small-scale human sampling to identify failure modes, designing specialized "separate prompts," and tuning them against human evaluation to achieve high accuracy.

### D.3 FAIL CASES FOR UNIFIED PROMPTING STRATEGY

Here, we present two failure cases related to false premises. The first involves a question that contains a false premise, but the agent's answer fails to explicitly identify it.

> **Given question**: Is Yamoussoukro the capital of the East African country of Côte d'Ivoire?
>
> **Groundtruth answer**: Yes, but the Republic of Côte d'Ivoire is a West African country.
>
> **Candidate answer**: Yamoussoukro is the capital of Côte d'Ivoire, as confirmed by multiple sources, including Wikipedia and Britannica. However, Abidjan remains the de facto economic center and largest city.

**Judgment from unified prompting**: True

**Judgment from human**: False

The second involves a question that does not contain a false premise, but the agent incorrectly assumes one due to missing or unavailable evidence. Both types of cases sometimes pose challenges for LLM-based evaluators to accurately assess.

> **Given question**: What novels did the Prime Minister of Iceland in 1904 write during his lifetime?
>
> **Groundtruth answer**: The Prime Minister of Iceland in 1904, Hannes Þórður Hafstein, wrote the novel Brennivínshatturinn.
>
> **Candidate answer**: According to the search results, the name of the Prime Minister of Iceland in 1904 is not explicitly mentioned in the information provided (the link to Wikipedia does not return a specific name). In addition, all search results do not show that the Prime Minister of Iceland in 1904 had any literary experience. Therefore, the user's question may have a wrong premise - the name of the Prime Minister of Iceland in 1904 cannot be confirmed at present, and there is no evidence that he wrote a novel.
>
> **Judgment from unified prompting**: True
>
> **Judgment from human**: False

# E    MORE EXPERIMENT RESULTS

## E.1    SETUP

We evaluate a range of closed-source and open-source LLMs under our Agentic RAG framework, including including GPT-4o (`gpt-4o-2024-08-06`) (Hurst et al., 2024), o3-mini (`o3-mini-2025-01-31`) (OpenAI, 2025b), Claude-3.7-Sonnet (`claude-3-7-sonnet-20250219`) (Anthropic, 2025), DeepSeek-V3 (Liu et al., 2024), DeepSeek-R1 (Guo et al., 2025), Gemini-2.5-Flash (`gemini-2.5-flash`) (Gemini, 2025b), Gemini-2.5-Pro (`gemini-2.5-pro`) (Gemini, 2025c), Llama-4-Maverick-17B-128E-Instruct (Meta, 2025), and Qwen3-32B (Team, 2025b). All models are configured with a context window of 4096 tokens. For queries that exceed this limit, we truncate earlier turns in the conversation history. During web browsing, if a document exceeds the limit, we segment it into chunks of up to 4096 tokens, summarize each chunk independently, and then aggregate the summaries. Our Agentic RAG framework evolves from open-source projects like BabyAGI (yoheinakajima, 2025), AutoGPT (Significant-Gravitas, 2025), and KwaiAgent (Pan et al., 2023). Unless otherwise specified, our default search engine is DuckDuckGo. For the implementation of the search engine, we first use the API to retrieve results. If the API fails to return the correct results, we then use a web scraper to fetch the results. Our evaluated LLMs are all implemented via API calls with temperature equals to 1. For models with official deployments, we use the official APIs; for those without (*e.g.*, Llama-4-Maverick-17B-128E-Instruct), we rely on third-party hosted APIs. Under our typical experimental settings (maximum step of retrieval stage $T = 5$, maximum evidence-set length $n = 5$), each query roughly requires 36 API calls:

1. Retrieval stage: 5 calls for planning and action of the agent.

2. Augmentation stage: 1 call to extract relevant evidence.

3. Answer-generation stage: 5 calls for answering based on five evidence from the evidence set and one call for answering based on all the observations.

4. Evaluation stage: 12 calls (two LLMs) to evaluate six candidate answers.

Each query also consumes about 24k input tokens and produces roughly 4k output tokens. Most of the input tokens come from the retrieval and augmentation stages, since those involve lengthy interaction histories and, at times, reading very long external documents.

Due to the high resource consumption of multi-round interactions with the agent, we only used the full set of 1,032 data points for the main experiment in Table 3 which evaluates the performance of

different models. All other analyses were conducted on a smaller subset of 245 data points due to resource constraints. As a result, the performance of the same model may vary between Table 3 and the other tables.

Table 6: Performance of different LLMs on question attributes, measured by %. Time-Sen. denotes Time-Sensitive. Distr.Info. means Distracting Information. False Prem. stands for False Premise.

| Model | Multi-Hop | Long-Tail | Time-Sen. | Freshness | Distr. Info. | False Prem. |
|---|---|---|---|---|---|---|
| Llama-4-Maverick-17B-128E-Instruct | 9.04 | 9.04 | 12.35 | 10.53 | 9.59 | 20.00 |
| Qwen3-32B w/o think | 7.98 | 8.56 | 7.41 | 16.67 | 10.96 | 7.69 |
| Qwen3-32B w/ think | 10.64 | 8.56 | 10.49 | 10.66 | 8.22 | 19.23 |
| DeepSeek-V3 | 6.38 | 8.02 | 9.26 | 12.50 | 12.33 | 19.23 |
| DeepSeek-R1 | 14.89 | 12.30 | 16.05 | 16.67 | 13.70 | 30.77 |
| GPT-4o | 6.91 | 8.02 | 9.88 | 14.58 | 9.59 | 23.08 |
| o3-mini | 9.57 | 10.16 | 12.35 | 8.33 | 6.85 | 19.23 |
| Claude-3-7-Sonnet | 11.70 | 10.16 | 12.96 | 22.92 | 12.33 | 26.92 |
| Gemini-2.5-Flash | 12.77 | 13.37 | 13.58 | 20.83 | 9.59 | 26.92 |
| Gemini-2.5-Pro | 20.74 | 19.79 | 22.22 | 39.58 | 21.92 | 38.46 |

## E.2    MORE RESULTS FOR QUESTION ATTRIBUTES

This section presents the performance of the information seeking agent across different LLMs, search engines, and retrieval steps on various question attributes. The results for different LLMs are summarized in Table 6, for different search engines in Table 7, and for different retrieval step configurations in Table 8. From these results, we can draw several key conclusions:

LLM reasoning capabilities play a significant role in improving the agent's performance across multiple question attributes. Stronger reasoning models, such as DeepSeek-R1 and Gemini-2.5-Pro, show a marked improvement in answering both simple and complex question attributes compared to base models, suggesting that enhanced reasoning abilities allow the agent to better utilize retrieved evidence for more accurate answer generation.

Search engine quality also impacts the agent's performance, with Google and Yahoo outperforming other engines like DuckDuckGo and Bing in most cases (as shown in Table 7). This is consistent with the previous analysis, where search engines with better information coverage and relevance lead to higher accuracy. Models paired with high-quality search engines, especially for multi-hop or long-tail questions, consistently show better results.

Table 7: Performance of different search engines on question attributes, measured by %. Time-Sen. denotes Time-Sensitive. Distr.Info. means Distracting Information. False Prem. stands for False Premise.

| Model | Search Engine | Multi-Hop | Long-Tail | Time-Sen. | Freshness | Distr. Info. | False Prem. |
|---|---|---|---|---|---|---|---|
| Gemini-2.5-Flash | DuckDuckGo | 12.77 | 13.37 | 13.58 | 20.83 | 9.59 | 26.92 |
| | Bing | 31.91 | 31.55 | 34.57 | 37.50 | 24.66 | 34.62 |
| | Google | 32.98 | 33.16 | 38.27 | 39.58 | 24.66 | 42.31 |
| | Yahoo | 29.79 | 33.16 | 33.95 | 31.25 | 26.03 | 50.00 |
| DeepSeek-V3 | DuckDuckGo | 6.38 | 8.02 | 9.26 | 12.50 | 12.33 | 19.23 |
| | Bing | 15.43 | 18.18 | 18.52 | 16.67 | 16.67 | 38.46 |
| | Google | 25.53 | 29.41 | 24.69 | 27.08 | 31.51 | 26.92 |
| | Yahoo | 22.87 | 26.20 | 23.46 | 16.67 | 16.44 | 34.62 |

Increasing the number of retrieval steps ($T$) improves the agent's accuracy, with a noticeable enhancement in both ACC and IA@k as the maximum number of retrieval steps increases. This scaling effect highlights the agent's ability to refine its search and gather more evidence with additional computation time. However, the performance improvements for long-tail and distracting information questions are more limited, despite increasing the maximum number of retrieval steps. These types of questions are inherently more difficult to answer due to the sparse and noisy nature of the relevant information available on the web. As a result, even with more retrieval steps, the agent still struggles to effectively parse through irrelevant or misleading content.

In summary, both LLM reasoning capabilities and search engine quality have a profound impact on the agent's ability to accurately answer different types of questions. Increasing the retrieval steps provides noticeable improvements, particularly for simpler questions. However, long-tail and distracting information questions remain more challenging, indicating that better evidence filtering and improved retrieval strategies are crucial for handling these complex scenarios.

Table 8: Performance on question attributes under different maximum step $T$ limits, measured by %. Time-Sen. denotes Time-Sensitive. Distr.Info. means Distracting Information. False Prem. stands for False Premise.

| Model | Max Step | Multi-Hop | Long-Tail | Time-Sen. | Freshness | Distr. Info. | False Prem. |
|---|---|---|---|---|---|---|---|
| DeepSeek-V3 | 1 | 3.19 | 4.28 | 3.09 | 4.17 | 5.48 | 11.54 |
| | 3 | 7.45 | 6.95 | 9.88 | 16.67 | 8.22 | 15.38 |
| | 5 | 6.38 | 8.02 | 9.26 | 12.50 | 12.33 | 19.23 |
| | 10 | 11.17 | 8.56 | 13.58 | 22.92 | 14.67 | 26.92 |
| | 20 | 12.23 | 16.58 | 15.43 | 14.58 | 10.96 | 26.92 |
| Gemini-2.5-Flash | 1 | 4.79 | 8.02 | 7.41 | 6.25 | 5.48 | 30.77 |
| | 3 | 9.04 | 9.09 | 11.73 | 18.75 | 9.59 | 30.77 |
| | 5 | 12.77 | 13.37 | 13.58 | 20.83 | 9.59 | 26.92 |
| | 10 | 15.96 | 18.18 | 19.75 | 33.33 | 19.18 | 38.46 |
| | 20 | 20.74 | 20.32 | 23.46 | 37.50 | 20.55 | 38.46 |

### E.3 MORE RESULTS FOR TEST-TIME SCALING IN AGENTIC INFORMATION SEEKING

We conducted experiments to assess the performance of two models, DeepSeek-V3 and Gemini-2.5-Flash, as the maximum step $T$ in information seeking was increased from 1 to 20. The evaluation metrics include ACC, IA@k, EEU, and IC, and we present the results in Table 9. From these results, we can draw several key conclusions. Both models benefit from an increased number of steps, demonstrating that more retrieval actions lead to better accuracy and information relevance. Gemini-2.5-Flash performs better than DeepSeek-V3 at all retrieval rounds. For example, at 1 round, Gemini-2.5-Flash has an ACC of 7.35%, while DeepSeek-V3 has 4.49%. As the number of retrieval rounds increases, Gemini-2.5-Flash also sees significant improvement, with ACC reaching 22.86% at 20 rounds, outperforming DeepSeek-V3. EEU for both models increases with more retrieval rounds, reflecting a higher utility of the information retrieved. However, Gemini-2.5-Flash consistently shows a higher EEU compared to DeepSeek-V3, particularly at 3, 5, and 10 rounds. IC (Information Compactness) remains relatively stable for both models across different retrieval rounds, with Gemini-2.5-Flash maintaining a slightly better performance compared to DeepSeek-V3.

Table 9: Performance with varying maximum action rounds in the retrieval stage. ACC and IA@k are measured by %.

| Model | Max Turn | ACC | IA@1 | IA@2 | IA@3 | IA@4 | IA@5 | EEU | IC |
|---|---|---|---|---|---|---|---|---|---|
| DeepSeek-V3 | 1 | 4.49 | 4.08 | 2.86 | 4.08 | 4.08 | 4.08 | 0.909 | 4.052 |
| | 3 | 8.57 | 5.31 | 8.57 | 8.16 | 8.16 | 8.16 | 1.000 | 3.965 |
| | 5 | 8.98 | 5.71 | 7.35 | 9.39 | 9.39 | 10.20 | 1.136 | 3.926 |
| | 10 | 12.65 | 6.94 | 10.61 | 10.61 | 11.43 | 11.84 | 0.935 | 3.826 |
| | 20 | 15.92 | 11.43 | 12.65 | 13.88 | 15.51 | 15.10 | 0.974 | 3.759 |
| Gemini-2.5-Flash | 1 | 7.35 | 8.16 | 7.76 | 7.76 | 8.57 | 8.16 | 1.167 | 3.908 |
| | 3 | 11.84 | 13.88 | 13.47 | 14.69 | 14.29 | 13.88 | 1.241 | 3.771 |
| | 5 | 14.29 | 12.65 | 15.10 | 16.73 | 16.73 | 15.92 | 1.171 | 3.750 |
| | 10 | 19.59 | 15.92 | 17.96 | 20.41 | 20.00 | 20.41 | 1.042 | 3.602 |
| | 20 | 22.86 | 20.82 | 22.04 | 22.45 | 23.67 | 22.86 | 1.036 | 3.573 |

### E.4 MORE RESULTS AND POTENTIAL SOLUTIONS FOR RETRIEVAL INTERFERENCE

This section presents additional experimental results on the phenomenon of retrieval interference, where the retrieval of external information negatively impacts the model's ability to answer questions

correctly. Table 10 displays the results for open-source LLMs, Table 11 for closed-source LLMs, Table 12 for different search engines, and Table 13 for varying maximum retrieval steps.

Table 10: Interference rates of open-source LLMs, measured by %. Llama-4-Maverick denotes Llama-4-Maverick-17B-128E-Instruct. IR denotes interference rate, and DAR denotes Direct Answer Ratio.

| Model | IR | DAR |
|---|---|---|
| Llama-4-Maverick | 87.50 | 3.27 |
| Qwen3-32B w/o think | 100.00 | 3.67 |
| Qwen3-32B w/ think | 88.89 | 1.63 |
| Deepseek-V3 | 84.21 | 5.71 |
| Deepseek-R1 | 53.13 | 8.16 |

Table 11: Interference rates of closed-source LLMs, measured by %. IR denotes interference rate, and DAR denotes Direct Answer Ratio.

| Model | IR | DAR |
|---|---|---|
| GPT-4o | 61.54 | 5.31 |
| o3-mini | 61.11 | 7.35 |
| Claude-3-7-Sonnet | 58.33 | 4.90 |
| Gemini-2.5-Flash | 68.97 | 6.53 |
| Gemini-2.5-Pro | 60.34 | 9.39 |

Our experiments reveal that certain models are able to correctly answer some questions based solely on their internal knowledge. However, when these same questions are queried with online retrieval, the answers become incorrect, which we define as retrieval interference, where the additional information gathered from the web undermines the agent's initial response. To quantify the extent of this interference, we introduce the interference rate, which measures the proportion of questions that an LLM can answer correctly without retrieval but fails to answer correctly when web-based information retrieval is applied. Specifically, the interference rate is calculated as the fraction of questions that an LLM answers correctly without retrieval but answers incorrectly after retrieval, normalized by the total questions it initially answered correctly without retrieval.

Our findings across various open-source and closed-source LLMs, search engines, and retrieval max turns show that retrieval interference is a widespread issue, with interference rates ranging from 40% to 80%. This high interference rate significantly reduces the model's probability of answering questions correctly, as irrelevant or conflicting web content can override the model's confident internal knowledge.

Table 12: interference rates under different search engines, measured by %.

| Model | Search Engine | Interference Rate |
|---|---|---|
| Gemini-2.5-Flash | DuckDuckGo | 68.97 |
| | Bing | 50.00 |
| | Google | 46.87 |
| | Yahoo | 42.31 |
| DeepSeek-V3 | DuckDuckGo | 84.21 |
| | Bing | 84.21 |
| | Google | 53.33 |
| | Yahoo | 42.11 |

To mitigate this issue, several strategies can be considered:

- **Improving Model Confidence in Internal Knowledge**: One possible approach is to develop mechanisms that increase the model's confidence in its own accurate knowledge, reducing its tendency to override correct internal answers when external information contradicts it. This could involve enhancing the model's self-reflection capabilities or providing additional confidence scores for internally generated answers before querying external sources.

- **Better Evidence Filtering**: A more effective evidence selection mechanism can help minimize irrelevant or conflicting information. For example, the model could prioritize high-confidence sources or introduce a ranking mechanism that filters out low-quality, noisy, or contradictory web pages. Contextual relevance checks could also be incorporated to ensure that only information that aligns well with the query's context is used.

- **Knowledge Consistency Checks**: Implementing consistency checks between the retrieved evidence and the model's internal knowledge could further improve accuracy. If a retrieved document contradicts previously confirmed internal knowledge, the agent could either ignore the external information or flag it for additional verification before using it in the final answer generation.

- **Hybrid Retrieval and Reasoning Approaches**: A hybrid approach that combines retrieval-augmented reasoning with internal knowledge checks may help. For instance, the agent could first check its internal knowledge and retrieve only supplementary information when necessary, minimizing reliance on external sources. This would reduce the risk of introducing irrelevant information while still benefiting from dynamic search results when needed.

- **Search Engine Optimization**: Since certain search engines, such as Google and Yahoo, tend to return more relevant results, using a more efficient search engine for information retrieval may help reduce the chance of encountering conflicting or misleading data. Moreover, optimizing search queries to be more specific or context-aware could lead to more relevant results, thereby reducing retrieval interference.

Table 13: Interference rates under varying maximum step $T$, measured by %.

| Model | Max Step | Interference Rate |
|---|---|---|
| DeepSeek-V3 | 1 | 85.71 |
| | 3 | 80.00 |
| | 5 | 84.21 |
| | 10 | 53.33 |
| | 20 | 52.94 |
| Gemini-2.5-Flash | 1 | 73.33 |
| | 3 | 62.07 |
| | 5 | 68.97 |
| | 10 | 58.33 |
| | 20 | 65.52 |

The phenomenon of retrieval interference highlights a significant challenge in agentic information seeking tasks, where additional information retrieved from the web can degrade the model's performance. Our results suggest that improving the model's ability to confidently rely on internal knowledge, optimizing retrieval strategies, and employing better filtering mechanisms are crucial steps in mitigating this interference. Further research into these strategies could enhance the reliability and robustness of agentic RAG systems in real-world applications.

Table 14: Performance with different languages. ACC and IA@k are measured by %. Pred. Lang. denotes Predominant Language.

| Model | Language | ACC | IA@1 | IA@2 | IA@3 | IA@4 | IA@5 | EEU | IC |
|---|---|---|---|---|---|---|---|---|---|
| DeepSeek-V3 | Chinese | 8.98 | 5.71 | 7.35 | 9.39 | 9.39 | 10.20 | 1.136 | 3.926 |
| | English | 13.47 | 12.24 | 11.84 | 11.84 | 11.84 | 12.65 | 0.939 | 4.032 |
| | Pred. Lang. | 17.14 | 11.02 | 15.92 | 17.96 | 17.55 | 17.96 | 1.048 | 3.919 |
| GPT-4o | Chinese | 10.20 | 9.39 | 8.16 | 9.39 | 8.57 | 8.98 | 0.920 | 3.878 |
| | English | 11.02 | 8.16 | 9.80 | 10.61 | 11.02 | 11.84 | 1.074 | 3.889 |
| | Pred. Lang. | 14.69 | 12.65 | 12.24 | 11.43 | 12.24 | 12.24 | 0.861 | 3.870 |
| Gemini-2.5-Flash | Chinese | 14.29 | 12.65 | 15.10 | 16.73 | 16.73 | 15.92 | 1.171 | 3.750 |
| | English | 17.55 | 14.29 | 15.92 | 17.55 | 18.78 | 18.78 | 1.070 | 3.761 |
| | Pred. Lang. | 18.78 | 15.92 | 15.92 | 17.14 | 17.55 | 17.96 | 0.957 | 3.802 |

## E.5 DETAILS AND MORE RESULTS OF LANGUAGE IMPACT

We also investigate how different languages (e.g., Chinese, English, and each query's predominant language) affect an agent's information-seeking performance, with DuckDuckGo as the fixed

search engine. For the Chinese and English settings, we crafted both prompts and answers in the respective languages, and observed that the language of the search queries generated by the LLM closely matches the language of the prompt and question. As shown in Table 14, English queries substantially outperform Chinese ones. This is likely due to the broader coverage of English-language content and search tools. LLMs see far more English text during pre-training, so they're stronger at understanding and generating English search queries. Search engines index and rank English pages more comprehensively, yielding higher-quality results.

Table 15: Accuracy on question attributes with different languages, measured by %. Pred. Lang. denotes Predominant Language.

| Model | Language | Multi-Hop | Long-Tail | Time-Sen. | Freshness | Distr. Info. | False Prem. |
|---|---|---|---|---|---|---|---|
| DeepSeek-V3 | Chinese | 6.38 | 8.02 | 9.26 | 12.50 | 12.33 | 19.23 |
| | English | 10.11 | 12.83 | 12.96 | 16.67 | 10.96 | 30.77 |
| | Pred. Lang. | 14.89 | 14.97 | 14.81 | 20.83 | 15.79 | 36.00 |
| GPT-4o | Chinese | 6.91 | 8.02 | 9.88 | 14.58 | 9.59 | 23.08 |
| | English | 8.51 | 9.09 | 8.64 | 22.92 | 9.59 | 30.77 |
| | Pred. Lang. | 13.30 | 14.44 | 12.35 | 18.75 | 11.84 | 32.00 |
| Gemini-2.5-Flash | Chinese | 12.77 | 13.37 | 13.58 | 20.83 | 9.59 | 26.92 |
| | English | 16.49 | 15.51 | 14.20 | 25.00 | 19.74 | 32.00 |
| | Pred. Lang. | 18.62 | 17.11 | 17.28 | 20.83 | 15.79 | 28.00 |

For the predominant language setting, although we recorded each instance's predominant language in our dataset, it proved difficult to translate prompts and questions into every target language. Instead, we designed a language-aware prompt instructing the agent to search in its dominant language (prompt details provided below). Results in Table 14 show that this language-aware prompting yields the best overall performance, indicating that specifying the dominant language indeed helps the agent retrieve more relevant information online.

Furthermore, prompting for the dominant language yields larger improvements on models with weaker innate multilingual capabilities—such as DeepSeek-V3 and GPT-4o—which cannot autonomously switch their search language and thus require explicit prompt cues. By contrast, stronger multilingual models like Gemini-2.0-Flash generally auto-adapt their search language and depend less on prompt instructions, resulting in smaller gains from our language-aware prompting strategy.

Table 16: Language switching ratios across different LLMs, measured by %.

| Model | DeepSeek-V3 | Gemini-2.5-Flash | GPT-4o |
|---|---|---|---|
| Ratio | 26.12 | 41.63 | 19.59 |

We present the proportion of questions that involved language switching during the search process for the three models in Table 16. Here, "language switching" refers to using a language other than the original query language during search. As shown, the specific rate of language switching varies across different LLMs. Gemini exhibits a significantly higher switching rate, while DeepSeek and GPT-4o show lower rates. Given that English accounts for approximately 60% of queries, models should switch to niche languages when necessary to achieve better retrieval outcomes.

We also present the retrieval interference and answer accuracy of various question attributes under different language settings in Tables 17 and Table 15.

The language-aware prompt is as follows:

```
You are a {agent_name}, {agent_bio}, {agent_instructions}
Currently, you are in the task planning phase, where you will
be given a specific query to address. Please utilize LLM's
advantages and pursue efficient strategies for task planning.
1.  You have a short-term memory of approximately 4,000
characters.
2.  You do not require assistance or response from users.
```

Table 17: Retrieval interference under different languages, measured by %. Pred. Lang. denotes Predominant Language.

| Model | Language | Interference Rate |
|---|---|---|
| DeepSeek-V3 | Chinese | 84.21 |
| | English | 64.71 |
| | Pred. Lang. | 56.25 |
| GPT-4o | Chinese | 61.54 |
| | English | 65.00 |
| | Pred. Lang. | 52.94 |
| Gemini-2.5-Flash | Chinese | 68.97 |
| | English | 53.85 |
| | Pred. Lang. | 48.39 |

```
3.  You can use the reference tools mentioned when planning.

4.  Complex problems can be split into sub-problems and then
information can be collected, aggregated and authenticated.
Be sure to verify the truthfulness of the information.

5.  Stay humble and call the tool for questions you are
not sure about, but do not call the same tool with the same
parameters repeatedly.

6.  You can flexibly switch the language of the search term
to get more information.  You can choose to search in Chinese,
English, or the language related to the entity involved in
the question (for example, if the question involves a French
person, you can search in French)

7.  You can think and plan up to {max_iter_num} steps, so
strive to plan tasks as efficiently as possible.

8.  You have the capability for reflection and self-criticism;
reflect on past decisions and improve your strategies.

9.  If you have sufficient information to answer the given
query, invoke the termination tool to terminate planning.
Otherwise, continue planning new tasks while ensuring no
duplication with prior tasks.

{tool_specification}

{current_date_and_time}

{memory}

Given Query:  {query}

Based on the given question and existing tasks, plan a new
Task (no repetitions), and you can only generate the Task in
the following **JSON list** format:

[{
    "task_name": "task description",
    "command":{
        "name":"command name",
        "args":{
            "arg name":"value"
        }
    }
}]

Even if there is only one task or no task, it needs to be
returned in the form of a list.  Ensure that the Task can be
parsed by Python's json.loads function.
```

```
        If the completed Tasks are sufficient to answer the query,
        terminate planning.  Otherwise, create another Task that do
        not duplicate previous ones.
        A new Task:
```

### E.6   IMPACT OF ANSWER LLMS

In previous experiments, when evaluating a specific LLM, we use it across all stages of the pipeline, including retrieval, augmentation, generation, and $\phi(\cdot, \cdot)$ for computing ACC and IA@k. In this section, we explore different answer LLMs $\phi(\cdot, \cdot)$, where the information seeking and generation stages use different LLMs. The results are presented in Table 18. Here, the term "Original" denotes the scenario where the same LLM generates the answer for computing IA@k, while "Fixed" refers to using a fixed LLM, DeepSeek-V3, for answer generation when computing IA@k, regardless of the model used in the retrieval and augmentation stages.

Table 18: Performance of different answer LLMs for $\phi(\cdot, \cdot)$. "Original" denotes that the answer is generated by the same LLM used in the retrieval and augmentation stage, while "Fixed" means employing a fixed LLM, DeepSeek-V3, to generate an answer. ACC and IA@k are measured by %.

| Model | answer LLM | IA@1 | IA@2 | IA@3 | IA@4 | IA@5 | EEU | IC |
|---|---|---|---|---|---|---|---|---|
| Qwen3-32B w/ think | | 11.43 | 10.48 | 12.38 | 12.38 | 14.29 | 0.833 | 4.116 |
| DeepSeek-R1 | | 20.00 | 24.76 | 25.71 | 24.76 | 24.76 | 1.286 | 3.895 |
| GPT-4o | | 14.29 | 13.33 | 14.29 | 12.38 | 12.38 | 0.882 | 4.071 |
| o3-mini | Original | 18.10 | 17.14 | 17.14 | 18.10 | 18.10 | 1.056 | 3.875 |
| Claude-3-7-Sonnet | | 18.10 | 18.10 | 20.95 | 20.00 | 20.00 | 0.957 | 4.044 |
| Gemini-2.5-Flash | | 20.00 | 21.90 | 22.86 | 24.76 | 21.90 | 1.040 | 3.842 |
| Gemini-2.5-Pro | | 29.52 | 28.57 | 29.52 | 28.57 | 29.52 | 0.886 | 3.977 |
| Qwen3-32B w/ think | | 13.33 | 14.29 | 13.33 | 13.33 | 14.29 | 0.833 | 4.103 |
| DeepSeek-R1 | | 20.00 | 21.90 | 23.81 | 23.81 | 23.81 | 1.190 | 3.938 |
| GPT-4o | | 11.43 | 14.29 | 15.24 | 14.29 | 14.29 | 0.941 | 4.068 |
| o3-mini | Fixed | 18.10 | 17.14 | 17.14 | 17.14 | 18.10 | 1.056 | 3.910 |
| Claude-3-7-Sonnet | | 17.14 | 18.10 | 20.95 | 21.90 | 21.90 | 1.000 | 4.063 |
| Gemini-2.5-Flash | | 20.95 | 20.00 | 24.76 | 24.76 | 24.76 | 1.040 | 3.773 |
| Gemini-2.5-Pro | | 25.71 | 28.57 | 28.57 | 26.67 | 27.62 | 0.857 | 3.997 |

From the results in Table 18, we observe that the performance difference between the Original and Fixed configurations is relatively small. However, fixed LLM configurations generally perform slightly worse. This may be because the information seeking LLM is aware of the knowledge gaps it has and selects corresponding documents to serve as evidence. Doing so can compensate for any missing knowledge from the original LLM, resulting in higher answer accuracy and lower information redundancy. However, when switching to a different answer LLM in the fixed setup, this advantage is lost. The answer LLM might not possess the same domain-specific knowledge as the information seeking LLM, leading to inaccuracies in the final answer generation. This demonstrates that the alignment between the LLMs used for information seeking and answer generation plays a crucial role in achieving higher performance in agentic information seeking tasks.

### E.7   SUBSET ANALYSIS

Because some analyses were conducted on a subset, we performed a Distribution Similarity analysis to check for consistency between the subset and the full dataset. This involved comparing the subset against the full dataset across Attributes, Predominant Languages, and Source Counts. Additionally, we analyzed the performance of the subset and the full set across different attributes to ensure the observed trends persist on the full dataset.

**Language Distribution**   : As shown in the Table 19, the distribution of predominant languages (e.g., Chinese and English) and other niche languages is highly similar between the two datasets. (Languages with negligible proportions are omitted here.)

Table 19: Language Distribution.

| Language | Chinese | English | Japanese | Russian | Italian | Spanish | Korean | Arabic | French |
|---|---|---|---|---|---|---|---|---|---|
| Full Set | 66.15% | 63.55% | 5.79% | 3.95% | 3.86% | 2.60% | 2.22% | 1.58% | 1.68% |
| SubSet | 63.27% | 61.22% | 4.49% | 3.67% | 2.45% | 1.63% | 3.27% | 2.45% | 2.04% |

**Attribute Distribution** : The attribute distribution exhibits a similar Macro-trend: Multi-Hop and Long-Tail remain the dominant categories, followed by Time-Sensitive, while other attributes account for a lower proportion.

Table 20: Attribute Distribution.

| Attribute | Multi-Hop | Long-Tail | Time-Sen. | Freshness | Distr. Info. | False Prem. |
|---|---|---|---|---|---|---|
| Full Set | 83.82% | 81.88% | 68.31% | 4.94% | 15.41% | 4.85% |
| SubSet | 76.73% | 76.33% | 66.12% | 19.59% | 31.02% | 10.20% |

**Average Source Count** : The average counts for the full set and subset are 1.69 and 1.71, respectively, which are closely aligned.

To address the reviewer's core concern regarding "whether trends persist," we directly validated this by evaluating the success rates of all models across attributes on both datasets. From the above

Table 21: Accuracy on full set and subset.

| Model | Set | Multi-Hop | Long-Tail | Time-Sen. | Freshness | Distr. Info. | False Prem. |
|---|---|---|---|---|---|---|---|
| Deepseek-V3 | Full Set | 4.45% | 4.66% | 6.10% | 11.54% | 11.32% | 16.00% |
| | SubSet | 6.38% | 8.02% | 9.26% | 12.50% | 12.33% | 19.23% |
| Deepseek-R1 | Full Set | 8.84% | 8.37% | 8.79% | 15.38% | 10.81% | 23.40% |
| | SubSet | 14.89% | 12.30% | 16.05% | 16.67% | 13.70% | 30.77% |
| Qwen3-32B w/o think | Full Set | 5.21% | 4.98% | 4.47% | 15.38% | 8.18% | 14.00% |
| | SubSet | 7.98% | 8.56% | 7.41% | 16.67% | 10.96% | 7.69% |
| Qwen3-32B w/ think | Full Set | 8.14% | 7.04% | 6.91% | 9.62% | 5.66% | 12.00% |
| | SubSet | 10.64% | 8.56% | 10.49% | 10.66% | 8.22% | 19.23% |
| GPT-4o | Full Set | 5.21% | 5.42% | 6.22% | 13.46% | 7.55% | 14.00% |
| | SubSet | 6.91% | 8.02% | 9.88% | 14.58% | 9.59% | 23.08% |
| o3-mini | Full Set | 9.31% | 9.71% | 10.30% | 9.62% | 8.28% | 17.02% |
| | SubSet | 9.57% | 10.16% | 12.35% | 8.33% | 6.85% | 19.23% |
| Gemini-2.5-Flash | Full Set | 8.14% | 8.23% | 9.08% | 19.23% | 8.18% | 20.00% |
| | SubSet | 12.77% | 13.37% | 13.58% | 20.83% | 9.59% | 26.92% |
| Gemini-2.5-Pro | Full Set | 16.83% | 16.14% | 15.45% | 38.46% | 17.61% | 34.00% |
| | SubSet | 20.74% | 19.79% | 22.22% | 39.58% | 21.92% | 38.46% |

Table 21, we draw two key conclusions:

- **Model Gaps Persist**: Across both datasets, Gemini-2.5-Pro significantly outperforms all other models, while models like DeepSeek-V3 consistently perform at a lower level. Furthermore, reasoning (thinking) models consistently outperform non-reasoning models. This proves that the performance gaps observed on the subset are real and generalizable.

- **Attribute Trends Persist**: In both datasets, Freshness and False Premise are consistently the (relatively) "easiest" attributes, with all models achieving their highest scores here. Conversely, Multi-Hop, Long-Tail, and Distr. Info remain the most challenging. This aligns perfectly with our analysis in Figure 4.

### E.8 ERROR ANALYSIS

The failure modes of the agent, as well as the underlying reasons, can broadly be categorized into three scenarios:

**Correct Information Source, Incorrect Answer.** This occurs when the agent successfully retrieves relevant information but still provides an inaccurate or suboptimal answer.

- **Misinterpretation and Hallucination**: The LLM might misunderstand nuances, context, or key facts within retrieved documents, or ignore the correct answer and hallucinate a different one entirely.
- **Synthesis/Summarization Errors**: The agent cannot correctly integrate information from multiple sources, leading to omissions or contradictions.

**Insufficient Information Sources.** The agent's information-seeking process is flawed, leading to a lack of necessary information.

- **Under-Retrieval due to Capability Shortcomings**: The model cannot generate sufficiently good queries. This includes an inability to dynamically adjust queries based on context and poor query formulation, e.g., failing to break down multi-hop questions, or using vague/incorrect keywords.
- **Agent Overconfidence**: The agent prematurely terminates its search, either hallucinating that it already possesses the knowledge or wrongly believing a failed search was exhaustive. This can lead to critical information being missing.
- **Search Engine Limitations**: Even with the same search query, the quality of the search engine plays a significant role. A superior search engine might retrieve the correct information source, whereas a less capable one might fail to do so.
- **Evidence Extraction Failure**: The agent might encounter the correct information source during searching, but fail to extract it as evidence due to over-retrieval or hallucinated reasoning.

**Sufficient but Noisy Information.** The agent retrieves both right and noisy information, and noise negatively impacts the final answer.

- **Over-Retrieval**: Retrieving excessive information introduces irrelevant or distracting content that confuses the agent.
- **Hallucinated Reasoning due to Contradictory Information**: Faced with conflicting sources, the LLM may arbitrarily choose the wrong one, attempt to reconcile them incorrectly, or hallucinate a new "fact" to resolve the conflict.

### E.9 STABILITY OF EEU

To validate the stability of EEU, we conduct a variance analysis and confidence interval estimation for the EEU metric across all models using the Bootstrapping method. This provides a clear measure of variance and stability, making our conclusions more robust. From Table **??**, the analysis results indicate:

- **Strong Stability**: Although ACC does influence EEU, the Standard Deviation (SD) for the vast majority of models is controlled between 0.06  0.1. For a ratio metric, this variance is very small, indicating that the metric does not exhibit drastic fluctuations.
- **Bounded Confidence Intervals**: Even for models with lower ACC, the 95% confidence intervals remain Compact and Bounded. We did not observe any "anomalous explosion" of EEU caused by minor changes in ACC. The original EEU values we reported (Single Run) all fall within the 9% confidence intervals, proving that the figures reported in our paper are statistically reliable estimates.

### E.10 QUALITY OF AGENTIC CONSTRUCTION

First, we ensured the factual correctness and determinacy of the agentic data through Human Verification. At the final stage of the automated pipeline, we implemented a rigorous human review process to confirm that the agent-generated QA pairs are factually accurate and unambiguous. This ensures

Table 22: Standard Deviation (SD) and Confidence Interval (CI) for EEU.

| Model | EEU | SD | 95% CI |
|---|---|---|---|
| GPT-4o | 0.91 | 0.069 | [0.70, 0.97] |
| o3-mini | 1.11 | 0.066 | [1.02, 1.25] |
| Gemini-2.5-Flash | 1.22 | 0.096 | [1.09, 1.43] |
| Gemini-2.5-Pro | 1.28 | 0.074 | [1.13, 1.45] |
| DeepSeek-V3 | 1.18 | 0.100 | [0.98, 1.37] |
| DeepSeek-R1 | 1.28 | 0.065 | [1.09, 1.35] |
| Claude-3.7-Sonnet | 1.01 | 0.058 | [0.91, 1.13] |
| Qwen3-32B w/o think | 0.81 | 0.089 | [0.65, 0.97] |
| Qwen3-32B w/ think | 0.77 | 0.074 | [0.60, 0.89] |

Table 23: Compare between manual and agentic construction.

| Model | Manual Set | Auto Set | Trend / Consistency |
|---|---|---|---|
| Gemini-2.5-Pro | 20.83% | 16.07% | Consistent Top-1 (↓ Harder) |
| Claude-3.7-Sonnet | 13.61% | 11.61% | Consistent Top-tier |
| o3-mini | 12.50% | 10.12% | Consistent Mid-tier |
| DeepSeek-R1 | 11.37% | 8.54% | Consistent Mid-tier |
| Gemini-2.5-Flash | 13.33% | 7.14% | Consistent Drop |
| Qwen3-32B (Think) | 9.72% | 7.89% | Consistent Ranking |
| GPT-4o | 8.54% | 6.40% | Consistent Ranking |
| DeepSeek-V3 | 7.22% | 5.06% | Consistent Ranking |

that the agent-constructed set maintains the same high standard of data quality and factual accuracy as the manually constructed data, achieving a "Ground-Truth" level.

Second, we have added an "Empirical Comparison" in Appendix X to demonstrate quality in terms of difficulty. We split the dataset into a Manual Set (constructed by human experts) and an Auto Set (constructed by agents) and compared the performance (ACC) of mainstream models on these subsets. From the Table 23, we observed two key phenomena that strongly refute the hypothesis of bias:

- **Relative ranking is preserved**: Multiple representative models scored low on both sets, and their performance ranking remains highly consistent across the two subsets. This strongly proves that the Agent-constructed set is as challenging as the Human-constructed set and is consistent in the core capability of "distinguishing between strong and weak models."

- **Absolute scores are generally lower on the Auto set**: This indicates that the agent did not compromise question quality (e.g., by introducing simple or biased shortcuts). Instead, it demonstrates that our automated framework's ability to mine Hard Questions effectively—potentially even surpassing human annotators. This heightened difficulty is precisely the contribution we aimed to introduce through automation, rather than a harmful bias.

## F  BROADER IMPACTS

The work presented in this paper, specifically the development of the InfoDeepSeek benchmark for agentic information seeking tasks, has several potential positive societal impacts. By improving the ability of language models (LLMs) to accurately retrieve and synthesize information, this research can enhance various applications such as virtual assistants, educational tools, and decision-support systems, making them more reliable and efficient. These improvements could contribute to advancing fields such as healthcare, law, and research by providing accurate, up-to-date, and contextually relevant information.

Our experiments reveal that current LLMs still exhibit significant shortcomings in agentic information seeking, exposing two primary areas of weakness: (1) the intrinsic reasoning and domain-knowledge

capabilities of the LLM itself, and (2) the quality and relevance of the search engine results it relies on. These findings carry both positive and cautionary implications:

- **Enhanced Reasoning Abilities of LLMs**: Stronger reasoning models (e.g., DeepSeek-R1, Gemini-2.5-Pro) consistently outperform baseline LLMs, pointing toward investment in specialized reasoning architectures.

- **Search Optimization**: Tailoring search queries and engines (as seen with Google/Yahoo gains) can substantially improve retrieval relevance. Future work might develop model-driven query rewriting or search-engine-specific adapters.

- **Long-Tail & Noise Handling**: Equipping agents with dedicated modules for identifying and filtering long-tail entities and distracting or conflicting information can reduce retrieval failures and improve focus.

- **Compute Scaling**: Allowing agents more compute at test time (i.e., increased retrieval steps) leads to clear scaling gains, suggesting that adaptive budgets or dynamic step policies could yield large benefits.

- **Mitigating Retrieval Interference**: Techniques such as internal-knowledge confidence checks or selective evidence fusion can prevent external noise from overriding correct model priors.

- **Language-Aware Retrieval**: Explicitly prompting agents to search in predominant languages unlocks richer, domain-specific resources, particularly for under-represented knowledge.

However, there are also negative societal impacts that must be considered. The advancements in LLMs and agentic RAG systems could potentially lead to misinformation amplification if the models are not properly evaluated or if they retrieve and generate content based on biased or misleading sources. Inaccurate or incomplete answers generated by models could exacerbate existing societal challenges, such as the spread of fake news or the reinforcement of harmful stereotypes. Additionally, as the technology becomes more powerful, there is the risk of misuse in areas like privacy violations or disinformation campaigns, where the system might be intentionally manipulated to produce harmful content.

