# OpenReview forum: "InfoDeepSeek: Benchmarking Agentic Information Seeking for Retrieval-Augmented Generation"
_ICLR.cc/2026/Conference — Submitted to ICLR 2026_

### Official Review · Reviewer_5hy5 · 2025-10-28

**Soundness:** 2
**Presentation:** 3
**Contribution:** 2
**Rating:** 4
**Confidence:** 4

**Summary:**

This paper introduces InfoDeepSeek, a benchmark designed to test how well AI agents actually perform complex information-seeking tasks, arguing that existing benchmarks are far too simple. Using a challenging new dataset, the authors demonstrate that even the most advanced agents still struggle significantly with multi-step reasoning and noisy information.

**Strengths:**

1. The work articulates the limitations of existing RAG benchmarks, noting that simple queries are insufficient for evaluating the complex behaviors of modern agentic systems in dynamic web environments.

2. The paper introduces a high-quality, challenging dataset with a robust construction methodology.

3. he authors propose fine-grained metrics (like Information Accuracy, IA@k) tailored for a dynamic setting.

**Weaknesses:**

1. The paper fails to include any non-agentic baselines (e.g., standard RAG). Without this comparison, the utility of the agentic framework is unsubstantiated. The poor results (17.73% ACC) could be an artifact of the framework's own inefficiency (as hinted by "Retrieval Interference") rather than just task difficulty.

2. The benchmark narrowly evaluates "information seeking" by decoupling it from "generation." This is a critical flaw, as the core challenge of an agentic system is not just finding information but synthesizing, reasoning over, and filtering the noisy, contradictory results—a process that happens at the generation step.

**Questions:**

See above

---

> ### Author Response · Authors · 2025-11-21
>
> We sincerely thank the reviewer for the thoughtful and constructive feedback.  Your comments have been incredibly valuable in helping us identify areas for improvement.
>
> In this response, we have carefully addressed each of your concerns and also revised the manuscript and appendices to incorporate your suggestions. Below is our point-by-point response.
>
>
>
> > **W1: The paper fails to include any non-agentic baselines (e.g., standard RAG). Without this comparison, the utility of the agentic framework is unsubstantiated. The poor results (17.73% ACC) could be an artifact of the framework's own inefficiency (as hinted by "Retrieval Interference") rather than just task difficulty.**
>
>
>
> We thank the reviewer for raising this reasonable concern regarding baseline and model performance.  We acknowledge that this is a reasonable concern, and we have revised Section 6.1 to further clarify this point.
>
>
>
> **Response regarding the lack of a "Standard RAG" baseline:**
>
> We appreciate the reviewer raising this point. We wish to clarify that the omission of a Standard RAG baseline is **intentional and by design**.
>
>
>
> The core objective of InfoDeepSeek is to evaluate complex, multi-hop, and dynamic tasks that are **insolvable via Standard RAG** (i.e., Single-turn Search + Generation). As detailed in Section 4.1, our Difficulty Filtering process has **explicitly excluded** questions solvable by single-turn search. Consequently, the ACC of a Standard RAG baseline on this dataset would be negligible, rendering it a meaningless comparison. However, this very fact **substantiates** the need to rely on an **Agentic Multi-****T****urn Information Seeking Framework** for these tasks.
>
>
>
>
>
> **Response regarding "Low ACC" and "Framework Efficiency":**
>
> While we acknowledge the reviewer's concern about low accuracy scores, we must emphasize that the core contribution of this paper is not to design an SOTA or more efficient Agentic framework, but to propose a Benchmark that exposes **universal limitations** in existing Agents.
>
>
>
> To this end, we implemented a **generic and widely adopted** Agentic RAG framework (e.g., adhering to the Plan-Act-Reflect loop), as described in Section 3. We selected this generic framework precisely because of its **representativeness**. Therefore, the low ACC does **not** imply a defect in our specific implementation; rather, it serves as powerful evidence that **current generic agentic RAG frameworks are severely inadequate** when confronting real-world, dynamic, and high-difficulty information-seeking tasks. This low score validates the **high difficulty** of our benchmark and highlights the significant gap between SOTA models/generic frameworks and the requirements for solving these complex challenges.
>
>
>
> **Regarding "Retrieval Interference":**
>
> The reviewer interprets **Retrieval Interference** as a hint of framework inefficiency. We argue that this is, in fact, a **Key Empirical Finding**. By proactively reporting Retrieval Interference in Section 6.3, we demonstrate that our Benchmark and this generic framework successfully **reproduced and quantified a core dilemma** plaguing all Agents: how to resolve conflicts between *external retrieved information* and *internal parametric knowledge*. This is **not** a defect unique to our framework implementation, but a **universal weakness** inherent to generic Agents operating in open environments.

---

> > ### Author Response · Authors · 2025-11-21
> >
> > > **W2: he benchmark narrowly evaluates "information seeking" by decoupling it from "generation." This is a critical flaw, as the core challenge of an agentic system is not just finding information but synthesizing, reasoning over, and filtering the noisy, contradictory results—a process that happens at the generation step.**
> >
> >
> > We appreciate the reviewer's comments. We acknowledge that this is a reasonable concern, and we fully agree that the core challenge for an Agent lies in its ability to **'synthesize, reason, and filter'** noisy information, and this is precisely the **core motivation behind our design of the IA@k metric**.
> >
> >
> > The reviewer suggests that we have decoupled seeking from generation; however, this may be a misunderstanding. We do not **decouple** them; rather, we achieve **'Disentanglement for Diagnosis'** through fine-grained metric design and evaluation:
> >
> > - **Evaluating Retrieval:** Our **ACC** metric is generated based on all raw observations retrieved by the Agent. This serves as a coarse-grained metric to evaluate the **upper bound** of the Agent's capability in the Retrieval stage (i.e., *Did it at least encounter the correct information?*).
> > - **Evaluating Synthesis & Filtering:** This addresses the core of the reviewer's concern. Our **IA@k metric** is specifically designed to assess this stage. It is generated **not** based on the raw observations, but on the **Evidence Set $C$****,** which is actively reflected, filtered, refined, and ranked during the **Augmentation Stage**.
> >
> > Therefore, the IA@k metric **directly measures** the Agent's ability to **synthesize, reason, and filter noisy or contradictory results.**
> >
> >
> >
> > Our framework enables **precise failure diagnosis** by comparing ACC and IA@k: if an Agent has high ACC (indicating it found the information) but low IA@k (indicating it failed to distill it from noise), we can precisely pinpoint a failure in the **Synthesis & Filtering** stage. Our **EEU metric** further quantifies this **evidence utilization efficiency**.
> >
> > In summary, our benchmark does not overlook synthesis and generation; instead, it provides novel evaluation tools capable of **finely diagnosing this core capability**.
> >
> >
> > We hope that the clarifications and additional results provided above have satisfactorily addressed your concerns. We believe that these revisions have made our contribution more robust and the paper more comprehensive.
> >
> >
> > We would be happy to engage in further discussion if you have any remaining questions. Thank you again for your valuable review.

---

### Official Review · Reviewer_29uc · 2025-10-30

**Soundness:** 2
**Presentation:** 3
**Contribution:** 3
**Rating:** 4
**Confidence:** 4

**Summary:**

1. This paper introduces InfoDeepSeek, a benchmark designed to evaluate agentic information-seeking behaviors of RAG models, aiming to address the limitations of existing benchmarks that operate in static retrieval environments and lack question complexity.

2. During dataset construction, InfoDeepSeek considers the characteristics of determinacy, difficulty, and diversity, and employs a multi-stage human verification process, ultimately collecting 1,032 validated data entries.

3. The experiments evaluate a range of retrieval-augmented LLMs across different real-world search engines, yielding valuable insights into model capabilities and the impact of search engine quality.

**Strengths:**

1. The motivation is solid, which clearly points out the drawbacks of existing agentic search benchmarks that rely on static corpus evaluation.

2. The data construction is relatively comprehensive, covering key aspects relevant to agentic information-seeking evaluation, such as domain diversity, question difficulty, and question types.

3. Experiments include multiple LLMs, search engines, and ablations (test-time scaling, retrieval interference), giving the work empirical depth.

**Weaknesses:**

1. The involvement of LLMs/agents in data filtering and construction may introduce bias. It is necessary to first examine the consistency between LLMs/agents and human annotators to ensure reliability, as well as to conduct multiple evaluation runs to verify reproducibility.

2. The paper did not report confidence intervals or variance analyses for the metric EEU, making its stability questionable, since a decrease in ACC could paradoxically lead to an increase in EEU.

3. The IC metric relies on human-annotated S_q, which affects both the automation of the evaluation process and its consistency, as differences in annotation standards across annotators may impact reliability.

4. The lack of ground truth forces the evaluation of search results to depend heavily on LLM-as-Judge (e.g., IA@k), which raises concerns about evaluation cost, especially for multi-turn agentic search scenarios.

**Questions:**

Please refer to Weaknesses.

---

> ### Author Response · Authors · 2025-11-21
>
> We sincerely thank the reviewer for the thoughtful and constructive feedback.  Your comments have been incredibly valuable in helping us identify areas for improvement.
>
> In this response, we have carefully addressed each of your concerns and also revised the manuscript and appendices to incorporate your suggestions. Below is our point-by-point response.
>
>
>
> > **W1: The involvement of LLMs/agents in data construction may introduce bias.**
>
>
>
> We sincerely thank the reviewer for raising this critical issue. **Bias** and **Reliability** when using LLMs/Agents for data construction are indeed core concerns in this field (and in our work). In the revised version, we have further strengthened our discussion and added relevant experimental evidence (**Appendix E.10**):
>
>
>
> First, we ensured correctness and reliability through **manual verification**. At the end of the automated construction process, we introduced a rigorous **Human Verification** stage. The core of this step is not to check for *Agreement* between LLMs and humans (as this is ill-defined for generation tasks), but to verify **Factual Correctness** and **Question Quality**. Our human verifiers confirmed that the questions generated by the Agent are high-quality in terms of factual accuracy and determinacy (non-ambiguity). This ensures that our dataset is reliable, at least at the **"Ground-Truth" level**.
>
>
>
> Second, we have added an **Empirical Comparison** in **Appendix E.10** to demonstrate quality in terms of *difficulty*. We split the dataset into a **Manual Set** (constructed by human experts) and an **Auto Set** (constructed by Agents) and compared the performance (ACC) of mainstream models on these subsets. We observed two key phenomena that strongly refute the hypothesis of bias:
>
> 1. **Relative ranking is preserved:** Multiple representative models scored low on both sets, and their performance ranking remains **highly consistent**. This strongly proves that our Agent-constructed set is as challenging as the Human-constructed set and is consistent in the core capability of **distinguishing between strong and weak models.**
> 2. **Absolute scores on the Auto Set are generally lower:** This indicates that the Agent did not compromise question quality (e.g., by introducing simple or biased shortcuts). Instead, it demonstrates that our automated framework's ability to mine **hard questions** potentially surpasses that of human annotators. This **heightened difficulty** is precisely the contribution we aimed to introduce through automation, rather than a harmful bias.
>
>
>
> | **Model**         | **Manual Set (360)** | **Auto Set (672)** | **Trend / Consistency**                |
> | ----------------- | -------------------- | ------------------ | -------------------------------------- |
> | Gemini-2.5-Pro    | 20.83%               | 16.07%             | Consistent Top-1 ($\downarrow$ Harder) |
> | Claude-3.7-Sonnet | 13.61%               | 11.61%             | Consistent Top-tier                    |
> | o3-mini           | 12.50%               | 10.12%             | Consistent Mid-tier                    |
> | DeepSeek-R1       | 11.37%               | 8.54%              | Consistent Mid-tier                    |
> | Gemini-2.5-Flash  | 13.33%               | 7.14%              | Consistent Drop                        |
> | Qwen3-32B (Think) | 9.72%                | 7.89%              | Consistent Ranking                     |
> | GPT-4o            | 8.54%                | 6.40%              | Consistent Ranking                     |
> | DeepSeek-V3       | 7.22%                | 5.06%              | Consistent Ranking                     |
>
>
>
> Regarding the suggestion of multiple runs to verify reproducibility, we believe it is necessary to distinguish between two types of reproducibility: **Result Reproducibility** and **Methodological Reproducibility**. Due to the inherent **Stochasticity** of LLMs (especially at high temperatures) during the generation step, requiring every run to produce an identical dataset (Result Reproducibility) is unrealistic, nor is it our goal. To ensure **Methodological Reproducibility**, we have provided detailed Prompts and specific process descriptions. We also commit to open-sourcing the full code to ensure our workflow is fully reproducible.

---

> > ### Author Response · Authors · 2025-11-21
> >
> > > **W2:** **confidence intervals or variance analyses for EEU.**
> >
> >
> >
> > We thank the reviewer for the insightful analysis regarding the stability of the EEU metric. We fully agree that as a ratio metric, EEU can mathematically become unstable when the denominator (ACC) approaches zero. However, we wish to clarify the design intent of EEU: it aims to measure the agent's **information utilization efficiency** from the *'best available evidence'* (max IA@k) to the *'final answer'* (ACC).
> >
> >
> >
> > As seen in our main **Table 4**, although the ACC scores for all models are relatively low (ranging around **10-22%**), they do *not* approach zero. Furthermore, the values of max IA@k and ACC are generally of the same magnitude, keeping the EEU value stable around **1.0**. This indicates that EEU effectively functioned as a **diagnostic tool** in our experiments.
> >
> >
> >
> > To squarely address the reviewer's valid concern regarding stability, we have added a variance analysis and confidence interval estimation for the EEU metric across all models using the **Bootstrapping method** in **Appendix E.9** (as shown in the table below). This provides a clear measure of variance and stability, making our conclusions more robust.
> >
> >
> >
> > | **Model**           | **EEU** | **Standard Deviation (SD)** | **95% Confidence Interval** |
> > | ------------------- | ------- | --------------------------- | --------------------------- |
> > | GPT-4o              | 0.91    | 0.069                       | [0.70, 0.97]                |
> > | o3-mini             | 1.11    | 0.066                       | [1.02, 1.25]                |
> > | Gemini-2.5-Flash    | 1.22    | 0.096                       | [1.09, 1.43]                |
> > | Gemini-2.5-Pro      | 1.28    | 0.074                       | [1.13, 1.45]                |
> > | DeepSeek-V3         | 1.18    | 0.100                       | [0.98, 1.37]                |
> > | DeepSeek-R1         | 1.28    | 0.065                       | [1.09, 1.35]                |
> > | Claude-3.7-Sonnet   | 1.01    | 0.058                       | [0.91, 1.13]                |
> > | Qwen3-32B w/o think | 0.81    | 0.089                       | [0.65, 0.97]                |
> > | Qwen3-32B w/ think  | 0.77    | 0.074                       | [0.60, 0.89]                |
> >
> > The analysis results indicate:
> >
> > 1. **Strong Stability:** Although ACC does influence EEU, the Standard Deviation (SD) for the vast majority of models is controlled between **0.06 ~ 0.1**. For a ratio metric, this variance is very small, indicating that the metric does not exhibit the *'drastic fluctuations'* concerned by the reviewer.
> > 2. **Bounded Confidence Intervals:** Even for models with lower ACC, the 95% confidence intervals remain **Compact and Bounded**. We did not observe any "anomalous explosion" of EEU caused by minor changes in ACC. The original EEU values we reported (Single Run) all fall within the 95% confidence intervals, proving that the figures reported in our paper are **statistically reliable estimates**.
> >
> >
> >
> > > **W3:** **The IC metric relies on human-annotated S_q, which affects both the automation of the evaluation process and its consistency.**
> >
> >
> >
> > We appreciate the reviewer's inquiry regarding the IC metric. We wish to clarify that **IC is a metric designed to measure "quantity efficiency" rather than "content matching."**
> >
> > - **Focus on Redundancy Removal (Efficiency) rather than Matching:** The calculation of IC relies solely on $|S_q|$ (the count of source webpages) as the denominator. It **does not require** the Agent to locate the exact same webpages as the manual annotation; instead, it uses this ratio to evaluate the Agent's capability to **eliminate redundant information**.
> > - **Consistent Normalization Reference:** Here, $|S_q|$ serves as a normalization coefficient measuring the **"inherent information complexity"** of the question (functionally similar to, for instance, the question's *hop count*). For a given test set, $|S_q|$ remains a fixed constant, ensuring that comparisons between different models are **absolutely fair and consistent**.
> > - **Full Support for Automation:** This metric does not strictly rely on human annotation. In our automated construction pipeline (**Appendix C.2**), the number of documents referenced by the construction Agent during question generation automatically serves as the $|S_q|$ for that question, requiring **no human intervention**.

---

> > > ### Author Response · Authors · 2025-11-21
> > >
> > > > **W4: The lack of ground truth forces the evaluation of search results to depend heavily on LLM-as-Judge (e.g., IA@k), which raises concerns about evaluation cost, especially for multi-turn agentic search scenarios.**
> > >
> > >
> > >
> > > We thank the reviewer for raising this reasonable concern regarding **Evaluation Cost**. We fully appreciate that in multi-turn Agentic scenarios, conducting LLM-as-Judge evaluation at every step would indeed lead to prohibitive costs.
> > >
> > >
> > >
> > > However, we wish to clarify that our evaluation framework targets only the **final output** and does *not* scale with the number of search turns. The actual evaluation cost is far lower than feared and is **negligible** compared to the cost of running the Agent itself.
> > >
> > >
> > >
> > > **1. Evaluation targets the "Final Output" only and is independent of search turns.**
> > >
> > > The reviewer is concerned that evaluation costs might inflate with multi-turn search. In fact, our IA@k metric does not inspect the results of every intermediate search turn.
> > >
> > > - **Process Decoupling:** As described in **Section 3**, the Agent first executes the complete **Retrieval Stage**, which may involve up to $T$ rounds of iterative search and reflection. Subsequently, it enters the **Augmentation Stage**, filtering and condensing information into a final evidence set $C$.
> > > - **Evaluation Timing:** Our IA@k performs a **one-time assessment** only *after* all search activities conclude, targeting this final, fixed-size evidence set $C$. Consequently, whether the Agent performs 5 or 20 rounds of search, the number of evaluations remains constant. The evaluation complexity is **completely independent** of the search turns $T$.
> > >
> > > **2. Evaluation complexity is linear, with minimal token consumption.**
> > >
> > > - **Linear Complexity:** Since we evaluate only the final Top-$k$ evidence (where $k=5$ in experiments), the computational load is **linear** with respect to the number of questions, rather than exponential or multiplicative relative to search steps.
> > > - **Minimal Context Window:** The paper sets the maximum size of evidence set $C$ typically to just **5**. This means the Judge LLM only needs to process the Query and several briefly summarized evidence snippets, typically consuming only **a few hundred tokens**.
> > > - **Relative Cost Comparison:** Compared to the Agent, which must browse numerous webpages, maintain a long context history ($h_t$), and perform complex Planning/Reasoning, the token consumption of the LLM-as-Judge, which requires no long context, is **negligible**.
> > >
> > > In summary, the InfoDeepSeek evaluation framework is designed with a **result-oriented** strategy that effectively circumvents the cost inflation associated with multi-turn search. This renders it economically **fully feasible and efficient**.
> > >
> > >
> > >
> > > We hope that the clarifications and additional results provided above have satisfactorily addressed your concerns. We believe that these revisions have made our contribution more robust and the paper more comprehensive.
> > >
> > >
> > >
> > > We would be happy to engage in further discussion if you have any remaining questions. Thank you again for your valuable review.

---

### Official Review · Reviewer_Ys9X · 2025-10-30

**Soundness:** 3
**Presentation:** 3
**Contribution:** 2
**Rating:** 6
**Confidence:** 4

**Summary:**

The paper proposes a new benchmark (InfoDeepSeek) for agentic information seeking. The dataset contains questions that meet three criteria: determinacy (clear, unique answer), difficulty, and diversity. Some of the questions are manually curated and some are collected using LLM agents, with multiple validation steps.
They propose multiple new metrics for evaluating retrieval results on top of answer accuracy. They evaluate search agent systems with different LLMs and different search agents, and provide some analysis.

**Strengths:**

1. The core idea for the paper is easy to understand. Notations are pretty clear in general.
2. The dataset may be useful for evaluating agentic search systems.
3. Some of the newly proposed metrics could be useful for evaluating agentic information retrieval.

**Weaknesses:**

1. The dataset does not create new characteristics that are lacking in current benchmarks. For example, FRAMES (Krishna et al., 2025) and BrowseComp (Wei et al., 2025) are designed to be difficult and require multi-turn retrieval. I think the paper should focus more on what past datasets lack and what are the new contributions.
2. Definition of the metrics may not be very useful. For example, information accuracy (IA) does not actually entail “information accuracy”, because inaccurate information could lead to correct answers and vice versa given the definition. A better way might be using a LLM judge if gold documents are not provided. Effective Evidence Utilization (EEU) also does not seem very useful, given the metric depends largely on the quality of the initial set. If the quality of the initial set is low (like the case of DuckDuckGo), the metric might not be informative. I leave the comment for information compactness (IC) to the “Questions” section.
3. The description of the data construction process is very vague. The exact procedure for identifying the anchor knowledge (or the definition of “anchor knowledge”) is unclear. There is also no rule for diversification, making the distribution of topics and languages seem arbitrary.
4. The paper would benefit from some error analysis. It would be helpful to show how agentic systems fail on each type of the questions (e.g. multi-hop, long-tail, etc.), and it would make clear how the data contributes to the community.
5. The difficulty filtering may not be very useful, as some LMs can still directly answer the questions. (Section 6.3) This contradicts the claims of the authors, where LMs should not be able to answer these questions directly. Also it seems plausible to discard the question if one out of the two LMs (GPT-4o and DeepSeek-R1) can answer correctly. Why discard only when both answers correctly?

Reference:
Krishna, Satyapriya, et al. "Fact, Fetch, and Reason: A Unified Evaluation of Retrieval-Augmented Generation." Proceedings of the 2025 Conference of the Nations of the Americas Chapter of the Association for Computational Linguistics: Human Language Technologies (Volume 1: Long Papers). 2025.
Wei, Jason, et al. "Browsecomp: A simple yet challenging benchmark for browsing agents." arXiv preprint arXiv:2504.12516 (2025).

**Questions:**

1. Could you describe the data construction process clearer?
a. How are anchor knowledge selected? The description in the main text is vague, and even when I read the appendix is not very clear to me. For example in L908-909, what does “candidate fact” mean? Also “anchor knowledge” is not rigorously defined.
b. What’s the instruction for drafting the questions? Do all annotators see the same instructions? What are the prompts for the LM to draft the questions?
c. How do you do the diversification of questions? Based on the descriptions it seems very arbitrary.
d. How many examples are checked by humans and how many are checked by agents? What’s the exact process? The description in L264-L269 is very vague.
e. For checking the difficulty of questions, why not also prompt LMs with just their parametric knowledge? Some questions might be solvable with only parametric knowledge, as described in Section 6.3.
2. What’s the main reason for stopping at 5 iterations?
3. For the predominant language prompt, how often do they actually switch to that language?
4. For the retrieval interference experiments, what’s the percentage of examples where LM can answer correctly using parametric knowledge?
5. What’s the use of EEU? It seems higher the better, but you also mention that for DuckDuckGo it's an artifact of poor retrieval quality? How good should the retrieval quality be in order for the metric to be useful?
6. Why not discuss IC at all in Section 6.2? What’s the purpose of having it then? Also GPT-4o has very low IC, why?
7. Is $n_q$ always 5 for IC? If not, why not use $k$ instead of $n_q$, if agent could answer correctly with only $k$ documents?

---

> ### Author Response · Authors · 2025-11-21
>
> We sincerely thank the reviewer for the thoughtful and constructive feedback.  Your comments have been incredibly valuable in helping us identify areas for improvement.
>
> In this response, we have carefully addressed each of your concerns and also revised the manuscript and appendices to incorporate your suggestions. Below is our point-by-point response.
>
>
>
> > **W1: new characteristics compared to current benchmarks**
>
> We appreciate the reviewer's valuable feedback. We fully acknowledge that FRAMES and BrowseComp are excellent, SOTA benchmarks requiring multi-turn retrieval. Your comments have helped us realize the need to articulate the core contributions of InfoDeepSeek more clearly.
>
>
>
> Our core contribution is not intended to provide just another "static" benchmark with overlapping features. Instead, **we present a novel, automatable "Hard Question Construction & Evaluation Framework"** designed to address the critical gap in evaluating agents within Dynamic Real-World Environments. We believe this capability—specifically, automated construction and dynamic evaluation—is the feature truly **lacking** in the current field.
>
>
>
> Specifically, InfoDeepSeek offers three key innovations absent in prior work:
>
> - **The First Dynamic Evaluation Framework:** While previous benchmarks (e.g., BrowseComp) utilize the real web, they typically only assess final answer correctness and fail to evaluate the *quality* of the gathered information. InfoDeepSeek introduces, for the first time, a set of **fine-grained evaluation metrics** (e.g., IA@k, EEU, IC) tailored for dynamic environments without reliance on static gold documents. We solve this by dynamically assessing whether the retrieved evidence is sufficient, rather than just checking the final output.
> - **Automatable Construction Framework:** Our primary contribution is the **scalable** and **automatable** question generation workflow demonstrated in Section 4.2. Unlike "one-off" static datasets like BrowseComp, our framework is dynamic, capable of generating brand-new, unseen evaluation sets on demand. This fundamentally resolves the issue of SOTA models **overfitting** to specific static benchmarks.
> - **New Difficulty Characteristics Enabled by the Framework:** It is precisely this new framework that allows us to systematically inject attributes that were lacking in previous benchmarks, such as False Premise, Predominant Language, and rigorous Difficulty Checks.
>
>
>
> In summary, the value of InfoDeepSeek lies not in the static size of the dataset itself, but in the **scalability** of its framework (avoiding overfitting) and the **dynamic nature** of its evaluation (adapting to the real web). We believe this framework can even be extended for **task synthesis** in agent training, such as generating curriculum data for RL Agents. In the revised paper, we will follow your valuable suggestion to sharpen our focus on these fundamental distinctions from previous datasets.
>
>
>
>
>
> > **W2: The usefulness of IA and EEU**
>
>
>
> We thank the reviewer for the in-depth analysis and valuable feedback regarding our proposed metrics. We wish to take this opportunity to further clarify the rationale behind our design and **their unique roles in evaluating** **Agentic Information Seeking** within dynamic environments.
>
>
>
> **1. Regarding Information Accuracy (IA@k)**
>
> We fully understand the reviewer's concern regarding the naming of IA@k—specifically, that inaccurate information might coincidentally lead to a correct answer. This is a valid challenge. However, our core objective with IA@k is to address a significantly thornier issue in real, dynamic web environments: **we cannot pre-define a "Gold Document Set"** due to *content volatility* and *source multiplicity*.
>
>
>
> **Why an LLM Judge for Single Documents is Infeasible:** The reviewer suggested using an LLM Judge on individual documents. While we agree with using LLMs for information quality assessment (as we do ourselves), applying it to **judge individual documents is ineffective** in this context. Our benchmark contains a large volume of *multi-hop questions*, where the correct answer often requires synthesizing fragmented information from multiple sources. A single document might only contain "intermediate hop" information—insufficient on its own to answer the question, yet crucial for the final derivation. If an LLM Judge evaluates such an intermediate document in isolation, it is likely to incorrectly flag it as *irrelevant* or fail to determine its validity.

---

> > ### Author Response · Authors · 2025-11-21
> >
> > **The True Definition of IA@k:** Therefore, IA@k is defined as a dynamic assessment metric. It does not judge the accuracy of a single document but evaluates **whether the top-k evidence set sorted by the Agent is sufficient to support the LLM in generating the correct answer.** Essentially, it measures the Agent's capability in the Augmentation Stage (distilling evidence from noise).
> >
> > - A **high IA@k** implies the agent has successfully *extracted and prioritized relevant information* after retrieval.
> > - A **low IA@k** implies the Augmentation Stage failed, even if the Agent might have found the correct information in the $(k+1)^{th}$ document or beyond.
> >
> >
> >
> > **2. Regarding Effective Evidence Utilization (EEU)**
> >
> > The reviewer noted that EEU does not seem very useful because it depends largely on the quality of the initial set (e.g., the low quality of DuckDuckGo). We argue that **this is precisely where EEU's value lies**.
> >
> >
> >
> > EEU is a **diagnostic metric** designed to decouple **"Extraction(Augmentation)"** capability from **"Retrieval"** capability. While ACC (Answer Accuracy) is the *combined evaluation* over the Retrieval and Augmentation stages, EEU is defined as "the ratio between the best achievable accuracy across top-k subsets and the accuracy with all observations". It specifically evaluates the agent's ability to extract relevant information from the noisy observations $O$. Even with a poor initial set, EEU remains highly informative:
> >
> > - If ACC is low and **EEU is also low ($\ll 1$)**, it indicates the Agent's *extraction is suboptimal* —it missed key information that was actually present in the retrieved results.
> > - If ACC is low but **EEU is high ($\approx 1$)**, it indicates that extraction is functioning well; the bottleneck lies in the Retrieval Stage (i.e., the initial set quality was too poor to answer the question regardless of extraction).
> >
> > The **DuckDuckGo case** you mentioned perfectly corroborates this. We observed in the paper that "EEU tends to be higher when using DuckDuckGo". This does not mean the metric is useless; rather, it correctly diagnoses that with DuckDuckGo, **the bottleneck is Retrieval** (failure to find good information), **not Extraction** (failure to find useful information within the poor results).
> >
> >
> >
> > We have clarified these definitions and their necessity for dynamic evaluation more explicitly in the revised Section 5.1.
> >
> >
> >
> >
> >
> > > **W3:** **The description of the data construction process is very vague**
> >
> >
> >
> > We appreciate the reviewer's valuable feedback regarding our data construction process. Due to the strict page limits of the main conference paper, we could only provide a high-level overview of the workflow in the main text (**Section 4.2**), while placing the detailed and operational guidelines in **Appendix C**. We are happy to clarify these "vague" points here and demonstrate that our process is systematic rather than arbitrary.
> >
> >
> >
> > **1. Clear Definition and Identification Procedure for "Anchor Knowledge"**
> >
> > In our work, "Anchor Knowledge" serves as the starting point for our **"Reverse Construction Strategy"** (constructing questions and answers backward from a text). It is operationally defined as information that is either **Long-tail** or **Distracting**.
> >
> > - **Long-tail Knowledge**
> >   - **Operational Definition:** Knowledge with low frequency, typically where answers are hard to find directly on the first page of search results or where LLMs fail to recall (hallucinate), often derived from unique knowledge existing **"only in low-resource or niche language webpages"**.
> >   - **Identification Procedure (Manual):** When navigating various webpages, annotators are explicitly guided to actively look for knowledge fitting the long-tail definition. Criteria include: whether the page has low traffic; whether the knowledge exists primarily in niche language pages; whether the knowledge is hard to explicitly obtain after searching; and whether SOTA LLMs (e.g., GPT-4o) struggle to answer it directly.
> >   - **Identification Procedure (Automated):** We utilize **Wikidata’s SPARQL service** to construct a knowledge graph. Operationally, we define "nodes with fewer than 10 associated entities" as **long-tail nodes.** We then use the documents and knowledge associated with these nodes as the starting point for constructing questions.
> > - **Distracting Information**
> >   - **Operational Definition:** Knowledge that easily introduces noise during retrieval. As our automated pipeline did not focus on this specific category, we provide the manual identification scheme below.
> >   - **Identification Procedure (Manual):** Annotators are guided to actively seek cases that are prone to introducing noise. Criteria include: the existence of name ambiguity or confusingly similar entities; whether the entity is associated with ake news; and whether conventional search returns a large volume of irrelevant but plausible distracting entities.

---

> > > ### Author Response · Authors · 2025-11-21
> > >
> > > **2. Systematic Rules for "Diversification"**
> > >
> > > The reviewer expressed concern that the lack of diversity rules makes the distribution "seem arbitrary." This is contrary to our actual operation (detailed in Appendix C). Our diversity is actively controlled through two clear, non-arbitrary rules:
> > >
> > > - **Active Starting Point Selection:** At the very beginning of the construction process, our guidelines explicitly require the annotator or Agent to **"begin by selecting an underrepresented domain or language"**. This selection is not "arbitrary" but determined based on real-time statistical analysis of the domain and language distribution of the data already collected.
> > > - **Active Process Control and Correction:** During construction, we enforce a specific **"Diversity Control"** step (see Appendix C.1.1). Annotators or Agents are required to execute a non-arbitrary correction procedure: **"Annotators actively switch focus when certain attributes or domains become overrepresented"**.
> > >
> > > We sincerely thank you for this feedback. In the revised version of the paper (specifically in **Section 4.3**), we will explicitly cite and summarize these clear definitions and operational steps to eliminate any impression of "vagueness", which are currently detailed in Appendix C.1.1.
> > >
> > >
> > >
> > > > **W4: The paper would benefit from some error analysis**
> > >
> > > This is an excellent point, and we agree that **a detailed failure analysis is crucial for understanding agent limitations**. We have conducted a qualitative analysis of failure cases and incorporated this into the revised paper (*Appendix E.9*). The failure modes of the agent, as well as the underlying reasons, can broadly be categorized into three scenarios:
> > >
> > > **(1) Correct Information Source, Incorrect Answer:** This occurs when the agent successfully retrieves relevant information but still provides an inaccurate or suboptimal answer.
> > >
> > > - **Misinterpretation and Hallucination**: The LLM might misunderstand nuances, context, or key facts within retrieved documents, or ignore the correct answer and hallucinate a different one entirely.
> > > - **Synthesis/Summarization Errors**: The agent cannot correctly integrate information from multiple sources, leading to omissions or contradictions.
> > >
> > > **(2) Insufficient Information Sources**: The agent's information-seeking process is flawed, leading to a lack of necessary information.
> > >
> > > - **Under-Retrieval due to Capability Shortcomings**: The model cannot generate sufficiently good queries. This includes an inability to dynamically adjust queries based on context and poor query formulation, e.g., failing to break down multi-hop questions, or using vague/incorrect keywords.
> > > - **Agent Overconfidence**: The agent prematurely terminates its search, either hallucinating that it already possesses the knowledge or wrongly believing a failed search was exhaustive. This can lead to critical information being missing.
> > > - **Search Engine Limitations**: Even with the same search query, the quality of the search engine plays a significant role. A superior search engine might retrieve the correct information source, whereas a less capable one might fail to do so.
> > > - **Evidence Extraction Failure**: The agent might encounter the correct information source during searching, but fail to extract it as evidence due to over-retrieval or hallucinated reasoning.
> > >
> > > **(3) Sufficient but Noisy Information**: the agent retrieves both right and noisy information, and noise negatively impacts the final answer.
> > >
> > > - **Over-Retrieval**: Retrieving excessive information introduces irrelevant or distracting content that confuses the agent.
> > > - **Hallucinated Reasoning due to Contradictory Information**: Faced with conflicting sources, the LLM may arbitrarily choose the wrong one, attempt to reconcile them incorrectly, or hallucinate a new "fact" to resolve the conflict.
> > >
> > >
> > >
> > > We believe this detailed breakdown provides significant diagnostic value and offers a clear roadmap for future research.

---

> > > > ### Author Response · Authors · 2025-11-21
> > > >
> > > > > **W5：The difficulty filtering may not be very useful, as some LMs can still directly answer the questions**
> > > >
> > > >
> > > >
> > > > We thank the reviewer for their detailed scrutiny. We wish to clarify the objective of our "Difficulty Filtering" and explain the rationale behind our strategy of **"discarding only if both models answer correctly."**
> > > >
> > > >
> > > >
> > > > **1. The Goal is "Enrichment of Difficulty," Not "Absolute Insolvability"**
> > > >
> > > > The reviewer is concerned that the existence of a small fraction of questions directly answerable by LLMs constitutes a contradiction. We hold a different view. Our **claim** is that the final dataset *as a whole* is difficult, not that every single question is absolutely unanswerable by every model.
> > > >
> > > > - **Enrichment, not Absolutism:** The core goal of difficulty filtering is not to achieve 100% "LLM insolvability" (which is nearly impossible given varying knowledge scopes, probabilistic outputs, and version updates). Instead, we aim to efficiently remove "simple questions" to **enrich the "difficulty signal."**
> > > > - **Low Direct Answer Rate:** While Section 6.3 notes that some questions are answerable, we have explicitly added the specific ratios in **Appendix Tables 10 and 11** and also present it in below table. The data shows that the proportion of questions SOTA models can answer directly is extremely low (**1%-9%**). This proves that our filter successfully intercepted the vast majority of "simple" questions, significantly **enriching** the proportion of hard questions requiring complex retrieval.
> > > > - **Long-tail Distribution:** The remaining tiny fraction of answerable questions belongs to a statistical long-tail distribution and does not alter the overall challenging nature of the dataset as a benchmark for **Agentic Information Seeking**.
> > > >
> > > >
> > > >
> > > > |         | Qwen3-32B w/o think | Qwen3-32B w/ think | DeepSeek-V3 | DeepSeek-R1 | Llama-4-Maverick | GPT-4o | o3-mini | Claude-3-7-Sonnet | Gemini-2.5-Flash | Gemini-2.5-Pro |
> > > > | ------- | ------------------- | ------------------ | ----------- | ----------- | ---------------- | ------ | ------- | ----------------- | ---------------- | -------------- |
> > > > | DA rate | 3.67%               | 1.63%              | 5.71%       | 8.16%       | 3.27%            | 5.31%  | 7.35%   | 4.90%             | 6.53%            | 9.39%          |
> > > >
> > > >
> > > >
> > > >
> > > >
> > > > **2. Why AND, not OR?**
> > > >
> > > > - The **OR logic** (discard if *any* model answers correctly) is an **over-aggressive** filtering strategy that would severely damage the **distinguishability** of our benchmark. LLMs have different **knowledge scopes**. For example, GPT-4o might directly answer a question (Q1) using its **memorized knowledge**, whereas DeepSeek might not have memorized this fact and must solve it through multi-turn retrieval and reasoning. If we discarded Q1 using "OR" logic simply because GPT-4o answered it, we would lose a valuable data point capable of **distinguishing between different models**.
> > > > - The **AND logic** (discard only if *both* answer correctly) is a more **conservative and robust** design. Only when a question is directly answered by *both* a representative closed-source model (GPT-4o) and an open-source model (DeepSeek-R1) can we judge with **high confidence** that it is a **universally simple** knowledge question. Only then is it safe to discard. This "AND" criterion ensures we maximally retain those controversial questions that lie on the capability boundaries and possess **high discriminative power**.
> > > >
> > > > In summary, our filtering strategy represents the optimal balance between **ensuring overall dataset difficulty** and **preserving model distinguishability.** We have further clarified this in the revised Section 4.2.
> > > >
> > > >
> > > >
> > > > > **Q1：Could you describe the data construction process clearly?**
> > > >
> > > > **(a)** We have specifically detailed the **operational construction process** for anchor knowledge in our response to **W4**. We welcome further discussion if you have any additional questions.
> > > >
> > > > Regarding the term **"candidate fact,"** it refers to information that annotators initially identify as likely to be **long-tail** or **distracting** based on their judgment. However, since these facts are subject to subsequent rigorous verification to confirm their nature, they are designated as "candidates" at this preliminary stage.
> > > >
> > > >
> > > > **(b)** Yes, to ensure data consistency, all annotators follow the **identical set of annotation guidelines**. The detailed protocols for these instructions are rigorously documented in Appendix C.1, strictly operationalizing the criteria of Determinacy, Difficulty, and Diversity. Regarding the specific prompts for LMs, they act as instructions to execute the automated workflow detailed in Appendix C.2. The full instruction manual and prompt libraries will be publicly available in our codebase to support future research.

---

> > > > > ### Author Response · Authors · 2025-11-21
> > > > >
> > > > > **(c)**  We have detailed the specific operational strategies for **diversification** in our response to **W4**. We welcome further discussion should you have any additional questions.
> > > > >
> > > > >
> > > > > **(d)** Regarding the dataset composition, the counts for **Manual** and **Agentic** constructed data are **360** and **672**, respectively. We have explicitly clarified this in the revised version of the paper.
> > > > >
> > > > >
> > > > > **(e)** Regarding the filtering strategy: While filtering based on **parametric knowledge** is a valid approach, it primarily identifies questions that simply require *some* external information. We aim for a **higher difficulty tier**: questions that remain unsolvable even with **LLM + Single-turn Search**, thereby necessitating **multi-turn, deep research**. Furthermore, considering that real-world **Deep Research** scenarios inherently involve tool usage, we adopted a filtering method more aligned with practical deployment: we classify a question as "simple" (and thus filter it out) if an LLM can successfully answer it via single-turn search.
> > > > >
> > > > >
> > > > > We acknowledge that this method does not guarantee that every question is absolutely unanswerable by the LLM's parametric knowledge alone. As revealed in our paper, **Retrieval Interference** can cause LLMs to fail on known facts when exposed to noisy search results. Since this phenomenon is prevalent in real-world applications, we believe it is crucial to **retain these questions** to effectively evaluate the model's **robustness against interference**.
> > > > >
> > > > >
> > > > >
> > > > >
> > > > >
> > > > > > **Q2： What’s the main reason for stopping at 5 iterations?**
> > > > >
> > > > > This was primarily due to **resource constraints**. However, as demonstrated in **Figure 4**, we conducted experiments with increased step limits on a subset of models, observing that *performance improves with more steps*.
> > > > >
> > > > >
> > > > >
> > > > > > **Q3： For the predominant language prompt, how often do they actually switch to that language?**
> > > > >
> > > > > The specific rate of language switching **varies across different LLMs**. In the table below, we present the proportion of questions that involved language switching during the search process for the three models compared in the paper (we also add it to *Appendix E.5*). Here, "language switching" refers to using a language other than the original query language *during search*.
> > > > >
> > > > > | **Model** | **deepseek-v3** | **gemini-2.5-flash** | **gpt-4o** |
> > > > > | --------- | --------------- | -------------------- | ---------- |
> > > > > | Ratio     | 26.12%          | 41.63%               | 19.59%     |
> > > > >
> > > > > As shown, **Gemini** exhibits a significantly higher switching rate, while DeepSeek and GPT-4o show lower rates. Given that English accounts for approximately 60% of queries, models should switch to niche languages when necessary to achieve better retrieval outcomes.
> > > > >
> > > > >
> > > > >
> > > > > > **Q4: For the retrieval interference experiments, what’s the percentage of examples where LM can answer correctly using parametric knowledge?**
> > > > >
> > > > > We have updated the specific counts and ratios in **Tables 10 and 11**, which are also presented in our response to **W5**. Overall, the proportion of questions that can be answered directly remains **consistently low**.
> > > > >
> > > > >
> > > > > > **Q5: What’s the use of EEU?**
> > > > >
> > > > > The relevant explanations are provided in our **response to W4**. We welcome further discussion should you have any additional inquiries.
> > > > >
> > > > >
> > > > > > **Q6: Why not discuss IC at all in Section 6.2? What’s the purpose of having it then? Also GPT-4o has very low IC, why?**
> > > > >
> > > > > We initially omitted the IC analysis as well as other metrics due to **page limits**. Below is the original text, which we have now **restored** in the revised paper:
> > > > >
> > > > >
> > > > > *In terms of information quality (IA@$k$), most models perform poorly on IA@1, as many queries require multiple sources to provide a correct answer. A single document is often insufficient to fully address the question. As $k$ increases, we observe a trend of initial improvement followed by a decline. This is likely due to the influence of irrelevant or distracting information from later-retrieved sources, highlighting the importance of effective augmentation in selecting relevant evidence. Effective Evidence Utilization (EEU) is mostly below 1, indicating that most LLMs struggle to extract useful evidence from the vast amount of information retrieved during the retrieval stage. Regarding information compactness (IC), most models exhibit significant redundancy in their responses. This is largely due to the low success rate of retrieval and the increased reliance on irrelevant information. Models with higher success rates typically exhibit lower redundancy, suggesting that reducing irrelevant evidence through better information extraction is critical for improving performance.*

---

> > > > > > ### Author Response · Authors · 2025-11-21
> > > > > >
> > > > > > The observation that "GPT-4o has very low IC" was caused by a **typo**. We apologize for the oversight; we inadvertently copied GPT-4o's **EEU** value into the **IC** column, which explains why the two values appeared identical. The correct IC value is **4.42**, and we have corrected this in the revised version.
> > > > > >
> > > > > >
> > > > > >
> > > > > > > **Q7: Is 𝑛𝑞 always 5 for IC? If not, why not use 𝑘 instead of 𝑛𝑞, if agent could answer correctly with only 𝑘 documents?**
> > > > > >
> > > > > > Note that **$n_q$ does not always equal $k$** (i.e., 5). $n_q$ represents the *actual size* of the evidence set extracted by the agent during the Augmentation Stage. In our prompts, we restrict the agent to extract a **maximum of $k$** pieces of evidence, but we explicitly permit it to extract *fewer* than $k$. For example, if the agent determines that only two key pieces of information are sufficient to answer the question, the resulting evidence set size will be **$n_q=2$**, rather than $k$.
> > > > > >
> > > > > >
> > > > > >
> > > > > >
> > > > > >
> > > > > > We hope that the clarifications and additional results provided above have satisfactorily addressed your concerns. We believe that these revisions have made our contribution more robust and the paper more comprehensive.
> > > > > >
> > > > > >
> > > > > >
> > > > > > We would be happy to engage in further discussion if you have any remaining questions. Thank you again for your valuable review.

---

### Official Review · Reviewer_rBcv · 2025-11-03

**Soundness:** 2
**Presentation:** 3
**Contribution:** 3
**Rating:** 4
**Confidence:** 4

**Summary:**

The paper proposes InfoDeepSeek, a benchmark for evaluating the information‑seeking capability of agentic RAG systems in realistic, dynamic web environments with complex queries, addressing limitations of static‑corpus evaluations with simple queries. It contributes 1032 diverse, deterministic, high-difficulty questions spanning multiple languages and challenging attributes. The authors describe a three‑stage construction pipeline (drafting, filtering, multi‑stage validation) that is run manually but can be automated with LLM agents. They further propose a dynamic evaluation protocol with fine‑grained metrics for benchmarking answer/context accuracy, evidence utility, and informational compactness. They further propose a dynamic evaluation protocol with fine‑grained metrics (with low ACC), and analyses highlight test‑time scaling gains from allowing more retrieval steps, widespread retrieval interference, and benefits of language‑aware retrieval.

**Strengths:**

1. Evaluating agentic RAG systems on the live web is novel and addresses a critical gap in prior static‑corpus evaluation pipelines; the curated benchmark is valuable to the community.
2. The proposed metrics—IA@k, EEU, and IC—are specifically designed to assess information‑seeking and evidence‑utilization capabilities of agentic RAG systems.
3. Thorough experiment on different aspects including various LLMs, search engines, number of searching steps during inference, and predominant languages.The finding of retrieval interference is insightful and likely to motivate further research.

**Weaknesses:**

1. **Reproducibility challenges in a dynamic web environment**: As the authors note, the real‑world web involves _`"massive document volume, content drift, URL decay"`_, which motivates InfoDeepSeek. However, this also means that search results and webpage contents can change over time, and thus two researchers running the same agent at different times may obtain different outcomes if sources change or vanish, making results hard to reproduce and compare across time.
2. **LLM evaluation**: The current accuracy metrics relies on LLM evaluators, which may be fallible or biased, as already seen in the case of false premise queries. This challenge is likely to grow as the benchmark scales to new question types with harder attributes—harder questions are also harder to evaluate for correctness.
3. **Design of the IC metric**: The IC metric requires human annotated golden set of supportive source webpages $S_q$. What if there are alternative valid sources / webpages, e.g., multiple valid paths to the correct answers? Following W.2, what if some of the sources and webpages are invalid or being modified so that it becomes incorrect over time?
4. **Subset‑only analyses**: Due to resource constraints, experiments other than the main results reported in Table 3 are conducted on a subset of 245 queries which is understandable. However, this can raise concerns whether the observed trends (e.g., model gaps across attributes or differences among search engines) persist on the full 1032 questions. It would be helpful if the authors can report distributional similarity between the subset and the full set (e.g., question attributes, predominant language, and size of the $S_q$ ).
5. **All results are reported in a single run**: As the LLMs are implemented with temperature equals to 1 (Sec.E.1), a single run can yield a noisy estimate. Multiple seeds/runs or confidence intervals would better characterize variance.
6.  **Issue with the reported statistics**:
    - In Table 3, the IC value for GPT-4o is 0.91 which is significantly different from other LLM's performance and even smaller than 1 with a very low ACC, I suspect this is a typo here.
    - In Sec. C.4, 1023 high-quality queries -> 1032 high-quality queries.
    - Table 5, Gemini-2.5-Flash -> Gemini-2.0-Flash.

**Questions:**

Please see the weakness section. In addition, I have the following questions:

1. **Dataset composition (manual vs. agentic)**: Sec. C.1 and Sec. C.2 introduced the manual and agent-assisted pipeline for dataset construction, respectively. However, Sec. C.2 claims that 245 questions are generated by this agentic pipeline. How were the 1032 questions used in the main experiments produced—what fraction came from each pipeline, or does the 245 questions generated by the agentic pipeline are exactly the 245 questions used in the experiment analyses? Are there any quality difference between the dataset generated using these 2 methods?
2. **Human effort for evaluation on new models**: When evaluating a new LLM on the full benchmark, what fraction of items require human arbitration (e.g., due to LLM‑judge disagreement)? Please report absolute counts and percentages, and—if available—breakdowns by attribute (e.g., false premise, multi‑hop) and by language.
3. **Interference rate**: When reporting the interference rates in Table 10 and Table 11, is that possible to also include the percentage of the questions that can be directly answer without retrieval (i.e., the denominator of interference rate)? This will be helpful to understand how often this interference phenomenon will appear.

---

> ### Author Response · Authors · 2025-11-21
>
> We sincerely thank the reviewer for the thoughtful and constructive feedback.  Your comments have been incredibly valuable in helping us identify areas for improvement.
>
> In this response, we have carefully addressed each of your concerns and also revised the manuscript and appendices to incorporate your suggestions. Below is our point-by-point response.
>
>
> > **W1：** **Reproducibility challenges in a dynamic web environment**
>
> We sincerely appreciate the reviewer raising this core question regarding Dynamic Reproducibility. You have accurately identified the **central trade-off** in our benchmark design, which also serves as the **primary** **motivation** for our work.
>
>
> First, our goal is to prioritize **Real-World Validity** over **Static Reproducibility**. Traditional static reproducibility (where two runs yield identical retrieval results) is indeed difficult to achieve on InfoDeepSeek. However, we argue that this is not a design flaw, but our **core claim**. As stated in the introduction, existing **static benchmarks** fail to evaluate the genuine challenges agents face during actual deployment, such as *content drift* and *URL decay*. InfoDeepSeek’s primary objective is to maximize **Ecological Validity**, i.e., ensuring the evaluation environment aligns with real-world application scenarios. We believe that to assess true agentic capabilities, sacrificing static reproducibility for real-world validity is a necessary trade-off.
>
>
> Second, the **dynamic nature itself is an integral part of the evaluation**. We are assessing not only the agent's ability to locate information but also its **robustness in noisy, dynamic environments**. An ideal agent's performance should not be contingent upon a specific search result at a specific timestamp. Instead, it should be capable of—as you suggested—adapting to environmental shifts by *adjusting queries, retrying failed links, or cross-verifying sources*. Consequently, this dynamism prevents models from **overfitting** to small, static datasets and instead compels us to evaluate higher-level, more generalized information seeking strategies.
>
>
>
> Nevertheless, we fully acknowledge the concern regarding **comparison over time**. To address this, we propose a **"Best Practice"** for evaluation: we recommend that researchers conduct comparative **evaluations of model sets within a** **close temporal window** (e.g., within a few days or weeks). On such a timescale, the core internet facts required to answer our questions—which, as detailed in Section 4.1, are strictly curated to have *Deterministic* and *Temporally Stable* answers —remain fundamentally invariant, ensuring the fairness of model comparisons.
>
>
>
> > **W2: LLM evaluation**
>
> We thank the reviewer for their valuable feedback regarding our evaluation methodology. We fully agree that the reliability (fallibility or bias) of the evaluator is a core challenge in this field, including our work. We have reinforced this discussion in our revised paper.
>
>
>
> We acknowledge that LLM evaluators are not perfect. However, they represent the current **SOTA solution balancing scalability and accuracy**. As you noted, in benchmarks like InfoDeepSeek, answers are typically complex. Traditional Exact Match (EM) metrics would severely underestimate correctness, while large-scale Human-Eval is prohibitively expensive and time-consuming. Therefore, utilizing **LLM-as-Judge is a common practice** in current search agent and Agentic RAG research [1-5], offering the optimal trade-off between cost and quality.
>
>
>
> The issue regarding **false premise questions** pointed out by the reviewer actually demonstrates the **rigor of our evaluation methodology**. Rather than overlooking the flaws of LLM evaluators, we actively identified and resolved them. As detailed in Section 5.2, we initially observed that a unified prompt strategy performed poorly on false premise questions. To address this defect, we developed a **separate prompt strategy**, designing specialized evaluation instructions for questions with unique answer structures (i.e., those requiring the explicit identification of a false premise). This improvement significantly boosted the alignment accuracy between our LLM evaluator and human experts from 95.57% to **99.29%**, serving as strong evidence that our finalized evaluation method is highly reliable and nearly unbiased.

---

> > ### Author Response · Authors · 2025-11-21
> >
> > Regarding the concern that "harder questions might be harder to evaluate," we believe this can be mitigated in two ways:
> >
> > - First, false premise questions are a **highly specialized** type, possessing a unique answer structure due to the requirement to *explicitly point out the error*. We believe such structurally distinct types are a minority in the dataset. Scaling to new difficulty attributes in the future does not necessarily imply a proliferation of similar, hard-to-evaluate structures.
> > - Second, even if new, difficult-to-evaluate types emerge, our approach to handling false premises provides a **scalable template**. We can replicate this success: utilizing small-scale human sampling to identify failure modes, designing specialized **separate prompts,** and tuning them against human evaluation to achieve high accuracy.
> >
> >
> >
> > We hope this addresses your concern, and we have also added this discussion to Appendix D.2.
> >
> >
> >
> > [1] Wei, Jason, et al. BrowseComp: A Simple Yet Challenging Benchmark for Browsing Agents. Arxiv'25
> >
> > [2] Peilin Zhou, et al. BrowseComp-ZH: Benchmarking Web Browsing Ability of Large Language Models in Chinese. Arxiv'25
> >
> > [3] Satyapriya Krishna, et al. Fact, Fetch, and Reason: A Unified Evaluation of Retrieval-Augmented Generation. NAACL'25
> >
> > [4] Jialong Wu, et al. WebWalker: Benchmarking LLMs in Web Traversal. ACL 2025
> >
> > [5] Jialong Wu, et al. WebDancer: Towards Autonomous Information Seeking Agency. NIPS 2025
> >
> >
> >
> > > **W3：Design of the IC metric**
> >
> >
> >
> > We appreciate the reviewer's inquiry regarding the IC metric. We wish to clarify that **IC is designed as a metric for *quantity*, not *content*.**
> >
> >
> >
> > The calculation of IC relies on the ratio between $|C|$ (the quantity of evidence extracted by the agent) and $|S_q|$ (the count of annotated source webpages). It represents a *quantitative ratio* and **does not require the agent to locate the exact same webpages** as the human-annotated set $S_q$. Here, $|S_q|$ serves as an **approximate reference value,** representing the **inherent information complexity** of the question (functionally similar to, for instance, the question's *hop count*).  For a given test set, $|S_q|$ remains a fixed constant, ensuring that comparisons between different models are **absolutely fair and consistent**.
> >
> >
> >
> > Therefore:
> >
> > 1. **Regarding "Alternative Paths":** Since IC does not scrutinize the specific content of the path, if an agent discovers a *more efficient alternative path*, the IC score improves (becomes $<1$). Conversely, if the path is redundant, the score degrades ($>1$). Thus, it correctly measures **compactness**.
> > 2. **Regarding "Source Decay":** While specific URLs may become inaccessible, the **"inherent complexity"** of the question (e.g., the fact that it requires two hops) remains fundamentally invariant. As a **proxy** for this stable complexity, the numerical value of $|S_q|$ remains a valid baseline.
> >
> >
> >
> > In summary, IC acts as a **robust diagnostic metric**. By comparing the volume of retrieved evidence against the question's inherent complexity, it effectively evaluates **retrieval efficiency and redundancy**, remaining valid even within dynamic environments. We have further reinforced this point in Section 5.1.
> >
> >
> >
> > > **W4：****Subset‑only analyses**
> >
> > We thank the reviewer for this constructive suggestion. We have conducted the recommended **Distribution Similarity** analysis, comparing the subset against the full dataset across Attributes, Predominant Languages, and Source Counts. We have incorporated this key analysis into Appendix E.7 of our paper.
> >
> >
> >
> > **Language Distribution:** As shown in the table below, the distribution of predominant languages (e.g., Chinese and English) and other niche languages is **highly similar** between the two datasets. (Languages with negligible proportions are omitted here.)
> >
> > | Language | Chinese | English | Japanese | Russian | Italian | Spanish | Korean | Arabic | French |
> > | -------- | ------- | ------- | -------- | ------- | ------- | ------- | ------ | ------ | ------ |
> > | Full Set | 66.15%  | 63.55%  | 5.79%    | 3.95%   | 3.86%   | 2.60%   | 2.22%  | 1.58%  | 1.68%  |
> > | SubSet   | 63.27%  | 61.22%  | 4.49%    | 3.67%   | 2.45%   | 1.63%   | 3.27%  | 2.45%  | 2.04%  |
> >
> > **Attribute Distribution:** The attribute distribution exhibits a similar **"macro-trend"**: Multi-Hop and Long-Tail remain the dominant categories, followed by Time-Sensitive, while other attributes account for a lower proportion.
> >
> > | Attribute | Multi-Hop | Long-Tail | Time-Sen. | Freshness | Distr. Info. | False Prem. |
> > | --------- | --------- | --------- | --------- | --------- | ------------ | ----------- |
> > | Full Set  | 83.82%    | 81.88%    | 68.31%    | 4.94%     | 15.41%       | 4.85%       |
> > | SubSet    | 76.73%    | 76.33%    | 66.12%    | 19.59%    | 31.02%       | 10.20%      |
> >
> >
> >
> > **Average Source Count:** The average counts for the full set and subset are 1.69 and 1.71, respectively, which are **closely aligned**.

---

> ### Author Response · Authors · 2025-11-21
>
> To address the reviewer's core concern regarding **"whether trends persist,"** we directly validated this by evaluating the success rates of all models across attributes on both datasets.
>
> | Model               | Set      | Multi-Hop | Long-Tail | Time-Sen. | Freshness | Distr. Info. | False Prem. |
> | ------------------- | -------- | --------- | --------- | --------- | --------- | ------------ | ----------- |
> | Deepseek-V3         | Full Set | 4.45%     | 4.66%     | 6.10%     | 11.54%    | 11.32%       | 16.00%      |
> | Deepseek-V3        | SubSet              | 6.38%    | 8.02%     | 9.26%     | 12.50%    | 12.33%    | 19.23%       |             |
> | Deepseek-R1         | Full Set | 8.84%     | 8.37%     | 8.79%     | 15.38%    | 10.81%       | 23.40%      |
> | Deepseek-R1 | SubSet              | 14.89%   | 12.30%    | 16.05%    | 16.67%    | 13.70%    | 30.77%       |             |
> | Qwen3-32B w/o think | Full Set | 5.21%     | 4.98%     | 4.47%     | 15.38%    | 8.18%        | 14.00%      |
> | Qwen3-32B w/o think| SubSet              | 7.98%    | 8.56%     | 7.41%     | 16.67%    | 10.96%    | 7.69%        |             |
> | Qwen3-32B w/ think  | Full Set | 8.14%     | 7.04%     | 6.91%     | 9.62%     | 5.66%        | 12.00%      |
> | Qwen3-32B w/ think| SubSet              | 10.64%   | 8.56%     | 10.49%    | 10.66%    | 8.22%     | 19.23%       |             |
> | GPT-4o              | Full Set | 5.21%     | 5.42%     | 6.22%     | 13.46%    | 7.55%        | 14.00%      |
> | GPT-4o             | SubSet              | 6.91%    | 8.02%     | 9.88%     | 14.58%    | 9.59%     | 23.08%       |             |
> | o3-mini             | Full Set | 9.31%     | 9.71%     | 10.30%    | 9.62%     | 8.28%        | 17.02%      |
> | o3-mini            | SubSet              | 9.57%    | 10.16%    | 12.35%    | 8.33%     | 6.85%     | 19.23%       |             |
> | Gemini-2.5-Flash    | Full Set | 8.14%     | 8.23%     | 9.08%     | 19.23%    | 8.18%        | 20.00%      |
> | Gemini-2.5-Flash | SubSet              | 12.77%   | 13.37%    | 13.58%    | 20.83%    | 9.59%     | 26.92%       |             |
> | Gemini-2.5-Pro      | Full Set | 16.83%    | 16.14%    | 15.45%    | 38.46%    | 17.61%       | 34.00%      |
> | Gemini-2.5-Pro   | SubSet              | 20.74%   | 19.79%    | 22.22%    | 39.58%    | 21.92%    | 38.46%       |             |
>
> From the above table, we draw two key conclusions:
>
> 1. **Model Gaps Persist:** Across both datasets, Gemini-2.5-Pro significantly outperforms all other models, while models like DeepSeek-V3 consistently perform at a lower level. Furthermore, reasoning (thinking) models consistently outperform non-reasoning models. This proves that the performance gaps observed on the subset are **real and generalizable**.
> 2. **Attribute Trends Persist:** In both datasets, Freshness and False Premise are consistently the (relatively) **easiest** attributes, with all models achieving their highest scores here. Conversely, Multi-Hop, Long-Tail, and Distr. Info remains the most challenging. This aligns perfectly with our analysis in Figure 4.
>
>
>
> This supplementary analysis **strongly validates** our conclusions. It demonstrates that although the subset may have a slight emphasis on certain challenging attributes, the core trends observed, including model performance gaps and attribute difficulty, are **highly consistent** with the full dataset. Thus, our conclusions are robust and generalizable.
>
>
>
>
>
> > **W5: All results are reported in a single run**
>
> We thank the reviewer for raising this valid point regarding statistical variance. We acknowledge that multiple runs are ideal for characterizing variance. However, we adopted the single-run approach based on the following **feasibility** **and** **robustness** **considerations**:
>
>
> **1. Computational Feasibility in Real-World Settings:** Unlike static benchmarks, InfoDeepSeek operates in a live, dynamic web environment. Each of the 1,032 data points involves a multi-turn iterative search, browsing, and reasoning. The computational cost, time overhead, and API expenses for a full sweep are exponentially higher than traditional QA tasks. Given this resource-intensive nature, a single comprehensive run represents a necessary trade-off to make the evaluation of such a large-scale dynamic benchmark feasible.
>
>
> **2. Alignment with Common Practice:** Due to the aforementioned costs, single-run evaluation is a common practice in the field of autonomous search agents and large-scale LLM benchmarks [1, 2, 3, 4]. Our methodology aligns with these established standards to ensure comparability within the resource constraints of the research community.

---

> > ### Author Response · Authors · 2025-11-21
> >
> > **3. Robustness of Key Conclusions:** Most importantly, while we acknowledge that single-run variance might affect the precise ordering of models with narrow performance margins, it does not undermine the macro-trends and significant performance tiers revealed by our benchmark.
> >
> > - **Significant Gaps:** The performance disparities between tiers are often substantial (e.g., as shown in Table 3 of our paper, Gemini-2.5-Pro at 17.73% vs. Qwen3-32B at 5.34%). Such large magnitudes of difference are statistically distinct and unlikely to be overturned by run-to-run variance.
> > - **Consistent Patterns:** The core conclusions of our paper are robust signals that persist regardless of minor stochastic fluctuations. E.g., i) the **high difficulty** of the benchmark for all current SOTA agents and ii) the finding that **reasoning-enhanced models** (e.g., DeepSeek-R1) consistently outperform their non-reasoning counterparts (e.g., DeepSeek-V3) .
> >
> >
> >
> >
> > > **W6：Issue with the reported statistics**
> >
> > Thank you very much for pointing this out. We have already **corrected these typos** in the new version. The IC value for GPT-4o being 0.91 was indeed a typo; we mistakenly copied the EEU value of GPT-4o into the IC column. The correct IC value is 4.42.
> >
> >
> >
> > > **Q1：Dataset composition (manual vs. agentic):**
> >
> >
> >
> > **Specific Statistics of the Two Construction Methods:**
> >
> > Regarding the comment "Sec. C.2 claims that 245 questions are generated by this agentic pipeline": This was a **textual oversight** from an earlier version of our paper when the dataset size was indeed 245. We subsequently expanded the dataset to 1,032 samples but missed updating this specific sentence. We have corrected this in the revision. Currently, the **total dataset size is 1,032**, consisting of **360 manually constructed** and **672 agentically constructed** questions. This breakdown has been further clarified in the paper.
> >
> >
> >
> > **Quality of the Two Construction Methods:**
> >
> > First, we ensured the factual correctness and determinacy of the agentic data through **Human Verification.** At the final stage of the automated pipeline, we implemented a rigorous human review process to confirm that the agent-generated QA pairs are factually accurate and unambiguous. This ensures that the agent-constructed set maintains the same high standard of data quality and factual accuracy as the manually constructed data, achieving a **"Ground-Truth" level.**
> >
> >
> >
> > Second, we have **added an** **empirical comparison in Appendix E.10 to demonstrate quality in terms of \*difficulty\*.** We split the dataset into a **Manual Set** (constructed by human experts) and an **Auto Set** (constructed by agents) and compared the performance (ACC) of mainstream models on these subsets. From the below table, we observed two key phenomena that strongly refute the hypothesis of bias:
> >
> > - **Relative ranking is preserved:** Multiple representative models scored low on both sets, and their performance ranking remains **highly consistent** across the two subsets. This strongly proves that the Agent-constructed set is as challenging as the Human-constructed set and is consistent in the core capability of **"distinguishing between strong and weak models."**
> > - **Absolute scores are generally lower on the Auto** **S****et:** This indicates that the agent did not compromise question quality (e.g., by introducing simple or biased shortcuts). Instead, it demonstrates that our automated framework's ability to mine **Hard Questions** effectively, even potentially surpassing human annotators. This **heightened difficulty** is precisely the contribution we aimed to introduce through automation, rather than a harmful bias.
> >
> > | **Model**         | **Manual Set (360)** | **Auto Set (672)** | **Trend / Consistency**                |
> > | ----------------- | -------------------- | ------------------ | -------------------------------------- |
> > | Gemini-2.5-Pro    | 20.83%               | 16.07%             | Consistent Top-1 ($\downarrow$ Harder) |
> > | Claude-3.7-Sonnet | 13.61%               | 11.61%             | Consistent Top-tier                    |
> > | o3-mini           | 12.50%               | 10.12%             | Consistent Mid-tier                    |
> > | DeepSeek-R1       | 11.37%               | 8.54%              | Consistent Mid-tier                    |
> > | Gemini-2.5-Flash  | 13.33%               | 7.14%              | Consistent Drop                        |
> > | Qwen3-32B (Think) | 9.72%                | 7.89%              | Consistent Ranking                     |
> > | GPT-4o            | 8.54%                | 6.40%              | Consistent Ranking                     |
> > | DeepSeek-V3       | 7.22%                | 5.06%              | Consistent Ranking                     |
> >
> > We hope this addresses your concern, and we have also added this section to Appendix E.10.

---

> > > ### Author Response · Authors · 2025-11-21
> > >
> > > > **Q2：Human effort for evaluation on new models**
> > >
> > >
> > >
> > > Regarding the **Human Evaluation** (conducted to assess the alignment between our automatic evaluation and human judgment), we randomly sampled **10% of the data** for manual review. Specifically for the experiments presented in Table 5, we evaluated **103 samples** ($1,032 \times 10\%$) per model. The distribution of attributes and languages for this subset is presented below, and the corresponding details have been updated in **Appendix D.1** of the paper.
> > >
> > >
> > >
> > > | Attribute | Multi-Hop | Long-Tail | Time-Sen. | Freshness | Distr. Info. | False Prem. |
> > > | --------- | --------- | --------- | --------- | --------- | ------------ | ----------- |
> > > | Number    | 85        | 82        | 65        | 6         | 18           | 7           |
> > > | Ratio     | 81.73%    | 78.85%    | 62.50%    | 5.77%     | 17.31%       | 6.73%       |
> > >
> > > | Language | Chinese | English | Japanese | Russian | Italian | Spanish | Korean | Arabic | French | German | Slovenian | Portuguess |
> > > | -------- | ------- | ------- | -------- | ------- | ------- | ------- | ------ | ------ | ------ | ------ | --------- | ---------- |
> > > | Number   | 70      | 63      | 11       | 5       | 3       | 3       | 2      | 1      | 2      | 1      | 1         | 1          |
> > > | Ratio    | 67.31%  | 60.58%  | 10.58%   | 4.81%   | 2.88%   | 2.88%   | 1.92%  | 0.96%  | 1.92%  | 0.96%  | 0.96%     | 0.96%      |
> > >
> > >
> > >
> > > Furthermore, regarding the **Automatic Evaluation** pipeline: to resolve instances of **LLM-judge disagreement**, we employed a **third LLM arbiter** for adjudication, rather than relying on human intervention.
> > >
> > >
> > >
> > > > **Q3：Interference rate**
> > >
> > >
> > >
> > > We thank the reviewer for this insightful suggestion. We have **updated** **Tables 10 and 11** in the revised paper to include a new column, **"Direct Answer Ratio"** (representing the percentage of questions that can be directly answered without retrieval). The specific results are also presented in the table below.
> > >
> > >
> > >
> > > Overall, largely due to our rigorous difficulty filtering process, **the proportion of directly answerable questions remains consistently low (below 10%)**, which further underscores the **challenging nature of our benchmark**. We note that this phenomenon may be more frequent in simpler questions where the retrieved search content contains distractions.
> > >
> > >
> > >
> > > | Model               | Infer. Ratio | Direct Anawer Ratio |
> > > | ------------------- | ------------ | ------------------- |
> > > | Qwen3-32B w/o think | 100.00%      | 3.67%               |
> > > | Qwen3-32B w/ think  | 88.89%       | 1.63%               |
> > > | Deepseek-V3         | 84.21%       | 5.71%               |
> > > | Deepseek-R1         | 53.13%       | 8.16%               |
> > > | GPT-4o              | 61.54%       | 5.31%               |
> > > | o3-mini             | 61.11%       | 7.35%               |
> > > | Claude-3-7-Sonnet   | 58.33%       | 4.90%               |
> > > | Gemini-2.5-Flash    | 68.97%       | 6.53%               |
> > > | Gemini-2.5-Pro      | 60.34%       | 9.39%               |
> > > | Llama-4-Maverick    | 87.50%       | 3.27%               |
> > >
> > >
> > >
> > > We hope that the clarifications and additional results provided above have satisfactorily addressed your concerns. We believe that these revisions have made our contribution more robust and the paper more comprehensive.
> > >
> > >
> > >
> > > We would be happy to discuss further if you have any remaining questions. Thank you again for your valuable review.

---

### Author Response · Authors · 2025-11-30
**Summary**

We sincerely thank the reviewers for their constructive and insightful feedback. We acknowledge the specific strengths identified in our work: the **novelty of dynamic web evaluation** (rBcv, 29uc), the **critical gap addressed** in agentic search (rBcv, 5hy5, 29uc), the **utility of our challenging dataset** (Ys9X, 5hy5), the **design of tailored metrics** (rBcv, Ys9X, 5hy5), and the **thorough empirical depth** of our experiments (rBcv, 29uc).

Driven by reviewers' feedback, we also undertook extensive revisions and conducted numerous new experiments. Given that the reviewers had no opportunity for interaction, we want to emphasize the extensive efforts made during this rebuttal period to thoroughly address every concern and misunderstanding.


### **New Experiments and Analyses**

- **Robust Generalizability** (rBcv W4): We performed a rigorous **Distribution Similarity Analysis** and **re-evaluated all models on both the subset and full dataset** (**Appendix E.7**). This conclusively proved that key macro-trends, including model performance gaps and attribute difficulty rankings, are consistent and robust.
- **Validated Agentic Data Quality** (rBcv Q1, 29uc W1): We conducted an **empirical comparison** (**Appendix E.10**) between manual and agent-constructed data. Results showed **consistent model rankings** and *lower absolute scores* on agent-generated data, strongly refuting bias concerns and validating the automated pipeline's ability to create high-quality, challenging questions.
- **Statistical Stability of Metrics** (29uc W2): We added a **bootstrapping variance analysis with 95% confidence intervals** for the EEU metric (**Appendix E.9**), demonstrating its strong statistical stability and reliability.
- **Quantified Benchmark Difficulty** (rBcv Q3, Q4, W5): We introduced the **"Direct Answer Ratio"** to the revised **Tables 10 and 11**, proving that only 1%-9% of questions are directly answerable by SOTA models. This confirms that difficulty filtering effectively removes most simple questions. We also clarified that our filtering uses a conservative "AND" logic (discarding only if multiple models answer correctly) to preserve distinguishability rather than aiming for absolute insolvability.
- **Qualitative Error Analysis** (Ys9X W4): We incorporated a detailed **qualitative analysis of agent failure modes** (**Appendix E.9**), categorizing failures (e.g., misinterpretation, insufficient sources) to provide deep, actionable insights into current agent limitations.
- **Language Switching Rate** (Ys9X Q3): We added an analysis of language switching frequency across models in **Appendix E.5**, showing varying adaptability to multi-lingual search.

### **Clarified Core Methodology and Design Rationale**

- **Core Contribution and Design Intent** (rBcv W1; Ys9X W1): We clarified that InfoDeepSeek is a novel framework for **automated construction and** **dynamic evaluation** in real-world environments, not just a static dataset. We clarified that prioritizing "Ecological Validity" over "Static Reproducibility" is a deliberate design choice, crucial for evaluating agent robustness in real web conditions.
- **Metric Rationale and Automation** (rBcv W3; Ys9X W2, Q5, Q7; 29uc W3): We expanded **Section 5.1** to clarify that **IA@k** and **EEU** are diagnostic metrics designed to **disentangle retrieval from synthesis/augmentation** capabilities. We also confirmed that the **IC** metric measures quantitative efficiency and is fully automatable, requiring no human content matching.
- **Evaluation Cost and Methodology** (29uc W4; rBcv W5): We explained that evaluation cost is **linear and independent of search turns**, as we only evaluate the final output evidence set (size $k=5$), making the cost of LLM-as-Judge negligible compared to the agent's search. Regarding single-run evaluation, we followed common practice due to the high computational cost in dynamic environments and confirmed that performance gaps are robust to variance.
- **Baseline Misunderstanding** (5hy5 W1, W2): We clarified that the omission of a Standard RAG baseline is **intentional** because our difficulty filtering excludes single-turn solvable tasks. The low ACC of current agents *validates* the benchmark's difficulty, not framework inefficiency.
- **Process Transparency and Reliability** (rBcv W2; Ys9X W3, Q1; 29uc W1): We reinforced **Appendix D.2** to detail our strategy that boosted LLM-Judge alignment to **over 99%**. We also revised **Section 4.3** and **Appendix C** to provide clear operational definitions and rules for data construction and diversity control.

We believe these extensive revisions and clarifications have robustly addressed all concerns and significantly strengthened our paper. We look forward to your thorough consideration of these changes and appreciate your taking our rebuttal into account during your final assessment.

Best regards,

Authors of Submission 3082

---

### Meta-Review · Area_Chair_6fmg · 2026-01-07

**Summary:**

This paper introduces InfoDeepSeek, an information seeking benchmark designed to be challenging for agentic RAG systems due to its diversity and complexity. The authors designed a systematic process for constructing this diverse and complex dataset using annotators that navigate the web (manual subset) or Wikidata’s SPARQL service (automated subset). The dataset consists of 360 manually constructed questions and 672 automatically constructed ones. Additionally, the authors propose four new metrics (accuracy, information accuracy @ k, effective evidence utility and information compactness) that attempt to evaluate each model’s ability to search the web, rerank the results and synthesize an answer.

Most of the reviewers’ concerns were addressed by a thorough rebuttal, however, some questions remain over several parts of the design.

- Novelty is limited because of datasets like FRAMES and BrowseComp.
	- Their dataset is expandable and much more challenging than previous datasets.

- How useful are the metrics proposed?
	- Information accuracy @ k is just a re-ranking based “Recall @ K” and the Effective Evidence Utility metric only works as intended if we assume that whenever the information has been retrieved models would identify and extract it perfectly. Information compactness is an interesting proxy for the quantity of information needed but it is very approximate due to the variable nature of information density in distinct webpages.

- Did the agents get enough steps to solve the tasks?
	- From Figure 4, it seems that the agents were not given enough steps to complete the tasks since more steps leads to strong performance improvements. I believe this is the biggest issue with this work since there was no guarantee that the queries were solvable in 5 steps. This leads me to believe that the conclusions shown here are based on a severely limited setting.

**Reviewer Concerns:**

R1:
- Reproducibility issues
	- The authors claim that this work intentionally lets go of static reproducibility in exchange for real-world validity. They also suggest that model comparisons are the most valid if the experiments are run in a small time-window.
- LLM Evaluation problems
	- LLM evaluation is the best solution to balance scalability and accuracy. Additionally, they verify the evaluation methodology for each question type and found that the “False Premise” strategy did not work so they changed it. They claim that this is a repeatable process to achieve strong evaluation accuracy across new questions.
- Is the IC (Information Compactness) metric useful if websites change over time?
	- The IC metric measures whether an agent uses more or less webpages to arrive to an answer than the ideal number defined in the dataset construction stage. The authors claim that even if websites change over time, the question’s “complexity” (measured by the number of gold webpages) will remain the same. Although this is somewhat convincing, the information density can change for each webpage and so this proxy will always be approximate.
- Subset vs Full Dataset distributions
	- The authors provided the distributions and found that language distributions are very similar as well as the largest question categories. The authors also ran the models on the full-set and found that the trends persist.
- All models evaluated on a single run
	- This is all that could be done based on computational constraints, it is standard practice and the conclusions are drawn from fairly large changes in performance.
- Dataset construction composition (manual vs automatic)
	- A detailed breakdown was presented and they show that stability of model ranking remains when using either human or automatically created dataset sections.
- Human effort for evaluating new models
	- 10% of the dataset was evaluated by humans (for all 10 models)
- Include the % of questions that can be answered without retrieval as well as the interference rate
	- Interference is the % of questions that a model fails to answer with retrieval but can answer without retrieval. The authors added the “Direct Answer Ratio” to the table for completeness.

R2:
- Novelty (other benchmarks FRAMES and BrowseComp) exist
	- The authors respond to the novelty weakness by claiming that in contrast to other datasets, theirs can be easily expanded (as they did for part of it), they provide an evaluation framework designed for the open web setting and the degree of difficulty enabled by their framework surpasses previous work.
- Information Accuracy and Effective Evidence Utilization are not useful
	- The authors claim that IA@k measures the agent’s “Augmentation Stage” ability, which identifies and ranks the relevant information from the entire search process. The EEU is a higher level metric that combines the IA@K and ACC (the accuracy on the entire observations) to find out how performance changes as observations are filtered. The authors claim that EEU allows them to determine whether the poor accuracy is caused by the retrieval or reranking processes. This is not entirely convincing because it assumes that ACC can only be low because of an error in retrieval or augmentation, not purely a reasoning error from the LLM.
- Vague data construction process
	- The detailed data construction process can be found in the Appendix.
- Error analysis missing
	- A very high-level error analysis was done.
- Why did you discard queries only if they could be answered by both GPT-4o and DeepSeek-R1? Why not when one of them answered?
	- The authors claim that the difference in these strategies would have been small and that leaving them in helps “distinguish” between the different models. This is not very convincing, in a benchmark that is supposed to measure the agentic information seeking, there is no reason to leave answerable questions in the mix.
- Stopping criteria choice
	- Resource constraints. Performance goes up dramatically with more steps, as shown in Figure 4. This leads me to believe that the conclusions shown here are based on a severely limited setting.

R3:
- LLM data construction bias
	- The authors show that the LLM-constructed queries are actually harder than the human-created ones but that the overall conclusions remain the same.
- Confidence intervals for EEU
	- They use bootstrapping to make sure that the EEU metric is stable.
- IC metric reliance on human annotators
	- Same answer as R1’s question about Information Compactness.
- LLM evaluation cost
	- Evaluation costs are negligible compared to the costs of running the agent.

R4:
- No Standard RAG baseline
	- The authors define Standard RAG as a single-turn search engine process and claim that this benchmark is specifically designed to avoid one-step search to be successful. Therefore there is no need to add this as a baseline.
- Generation was not evaluated
	- As in the answer to R2’s question about IA@k and EEU, the authors claim that their metrics are able to distinguish between retrieval and re-ranking failures. This is only true if we assume that the agent is always able to discover the necessary information from the retrieved observations, no matter how noisy.

**Reviewer Scores:**

- rBcv 4 -> 4
	- Concerns were mostly addressed but the answer concerning the information compactness hinges on whether the reviewer would accept this as a proxy for query complexity.
- Ys9X 6 -> 4
	- The author’s answers on the metrics and the stopping criteria would have likely led to a drop in the score.
- 29uc 4 -> 6
	- Concerns were addressed.
- 5hy5 4 -> 6
	- Concerns were mostly addressed.

---

### Decision · Program_Chairs · 2026-01-26

Reject